# Multimodal MRI marker of cognition explains the association between cognition and mental health in the UK Biobank

Irina Buianova[1]*, Mateus Silvestrin[2,3], Jeremiah D Deng[4], Narun Pat[1]

[1]Department of Psychology, University of Otago, Dunedin, New Zealand; [2]Federal University of the São Francisco Valley, Petrolina, Brazil; [3]National Institute of Social and Affective Neuroscience, Petrolina, Brazil; [4]School of Computing, University of Otago, Dunedin, New Zealand

## eLife Assessment

This **valuable** work advances our understanding of the relationship between multimodal magnetic resonance imaging (MRI) measures, cognition, and mental health. **Compelling** use of statistical learning techniques in UK Biobank data shows that 48% of the variance between an 11-task derived g-factor and imaging data can be explained. Overall, this paper contributes to the study of brain-behaviour relations and will be of interest for both its methods and its findings on how much variance in g can be explained.

**\*For correspondence:**
is.buyanova@gmail.com

**Competing interest:** The authors declare that no competing interests exist.

**Abstract** Cognitive dysfunction often co-occurs with psychopathology. Advances in neuroimaging and machine learning have led to neural indicators that predict individual differences in cognition with reasonable performance. We examined whether these indicators explain the relationship between cognition and mental health in the UK Biobank ($n>14,000$). Using machine learning, we quantified the covariation between cognition and 133 mental health indices and derived neural indicators of cognition from 72 neuroimaging phenotypes across diffusion-weighted MRI (dwMRI), resting-state functional MRI (rsMRI), and structural MRI (sMRI). With commonality analyses, we investigated how much of the cognition–mental health covariation is captured by each indicator and neural indicators combined within and across MRI modalities. The predictive association between mental health and cognition was at $r=0.3$. Neuroimaging captured 2.1 to 25.8% of the cognition-mental health covariation. Combining phenotypes within modalities improved the explanation to 25.5% for dwMRI, 29.8% for rsMRI, and 31.6% for sMRI, and combining them across modalities enhanced the explanation to 48%. We present an integrated approach to derive multimodal MRI markers of cognition that can be transdiagnostically linked to psychopathology, demonstrating that the predictive ability of neural indicators extends beyond the prediction of cognition itself, enabling us to capture cognition-mental health covariation.

## Introduction

Cognition and mental health are closely intertwined (*Iosifescu, 2012*). Cognitive dysfunction is present in various mental illnesses, including anxiety (*Gulpers et al., 2022*; *Nyberg et al., 2021*), depression (*Kriesche et al., 2023*; *Richardson and Adams, 2018*; *Wen et al., 2022*), and psychotic disorders (*Chavez-Baldini et al., 2023*; *Fusar-Poli et al., 2012*; *Guo et al., 2019*; *Lindgren et al.,*

*2020*; *Mesholam-Gately et al., 2009*; *Semkovska et al., 2019*). National Institute of Mental Health's Research Domain Criteria (RDoC) (*Cuthbert and Insel, 2013*; *Insel et al., 2010*) treats cognition as one of the main basic functional domains that transdiagnostically underlie mental health. According to RDoC, mental health should be studied in relation to cognition, alongside other domains, such as negative and positive valence systems, arousal and regulatory systems, social processes, and sensorimotor functions. RDoC further emphasises that each domain, including cognition, should be investigated not only at the behavioural level but also through its neurobiological correlates. In this study, we aim to examine how the covariation between cognition and mental health is reflected in neural markers of cognition, as measured through multimodal neuroimaging.

Recent efforts in brain Magnetic Resonance Imaging (MRI) and machine learning have led to predictive models that allow us to create MRI-based neural indicators of cognition with reasonable predictive performance (*Krämer et al., 2024*; *Pat et al., 2022*; *Tetereva et al., 2022*). These models are designed to predict cognition based on different cognitive tasks in unseen individuals who are not part of the modeling process (*Marek et al., 2022*; *Zhi et al., 2024*). Yet, the extent to which MRI-based neural indicators designed to predict cognition capture the same variance that mental health shares with cognition remains unknown. Demonstrating that MRI-based neural indicators of cognition capture the covariation between cognition and mental health will thereby support the utility of such indicators for understanding the etiology of mental health (*Wang et al., 2025*).

Different MRI modalities measure different aspects of the brain, and MRI quantification techniques capture different brain features, resulting in distinct neuroimaging phenotypes. This means there are numerous approaches to derive neural indicators of cognition from MRI data. For example, diffusion-weighted MRI (dwMRI) measures the shape and amount of water diffusion in various directions and tissue compartments (*Alexander et al., 2007*). Different dwMRI metrics, such as fractional anisotropy (FA), which quantifies the degree of water diffusion directionality, and the streamline count, which indirectly reflects structural connectedness between the two regions (structural connectome), provide information about white matter orientation, density, and microstructural integrity (*Basser et al., 1994*; *Soares et al., 2013*; *Zhang et al., 2012*). Resting-state functional MRI (rsMRI) measures spontaneous low-frequency fluctuations in the Blood Oxygenation Level Dependent (BOLD) signal in the absence of a task, enabling the investigation of resting-state functional connectivity (RSFC) (*Lee et al., 2013*). RSFC from rsMRI can be estimated between pairs of parcellated grey matter regions (functional connectome) or between widespread networks derived from the Independent Component Analysis (ICA). Structural MRI (sMRI) uses T1-weighted and T2-weighted imaging to quantify various aspects of brain anatomy and morphology. For example, the morphology of the cerebral cortex and white matter can be quantified by measuring grey or white matter thickness, volume, and area in regions defined by different atlases, whereas the characteristics of subcortical regions are conventionally quantified with volumes of subcortical nuclei and their subdivisions (*Symms et al., 2004*; *Wattjes, 2011*). Previous studies using machine learning have shown that both (a) the choice of MRI modality and (b) the quantification method within each modality affect the performance of MRI-based models in capturing cognition (*Dhamala et al., 2021*; *Pat et al., 2022*; *Tetereva et al., 2022*). Dhamala and colleagues found that the predictive ability of structural and functional connectomes largely depends on the choice of atlases used to parcellate grey matter and how they were derived (*Dhamala et al., 2021*).

Given the heterogeneity of neuroimaging phenotypes from different MRI modalities, drawing information across them may boost the predictive ability of MRI-based neural indicators (*Caunca et al., 2021*). One way to integrate multiple neuroimaging phenotypes across MRI modalities is a stacking approach, which employs two levels of machine learning. First, researchers build a predictive model from each neuroimaging phenotype (e.g. cortical thickness from different grey matter parcellations) to predict a target variable (e.g. cognition). Next, in the stacking level, they use predicted values (i.e. cognition predicted from each neuroimaging phenotype) from the first level as features to predict the target variable (*Pat et al., 2022*). Previous studies show that integrating multimodal neuroimaging phenotypes into 'stacked models' enhances the prediction of cognition (*Krämer et al., 2024*; *Pat et al., 2022*; *Rasero et al., 2021*; *Tetereva et al., 2022*). Here, we aim to determine whether this improvement extends beyond the prediction of cognition itself, allowing us to capture more covariation between cognition and mental health.

**Table 1.** Characteristics of the train and test sets used to build the *g*-factor.

| | Number | | Age, Mean (SD) | | Females, % | |
|---|---|---|---|---|---|---|
| Fold | Train | Test | Train | Test | Train | Test |
| 1 | 25290 | 6323 | 64.52±7.64 | 64.48 (7.74) | 51.39 | 51.16 |
| 2 | 25291 | 6322 | 64.53±7.66 | 64.44 (7.62) | 51.19 | 51.96 |
| 3 | 25290 | 6323 | 64.53±7.66 | 64.45 (7.65) | 51.38 | 51.19 |
| 4 | 25290 | 6323 | 64.5±7.66 | 64.56 (7.65) | 51.27 | 51.64 |
| 5 | 25291 | 6322 | 64.48±7.67 | 64.63 (7.61) | 51.49 | 50.78 |

*SD*, standard deviation.

Using the largest population-level neuroimaging dataset, the UK Biobank, we investigated (a) which neuroimaging phenotypes yield a neural indicator of cognition that explains the relationship between cognition and mental health the most, and (b) whether combining neuroimaging phenotypes within and across MRI modalities enhances the explanation of this relationship. We started by deriving a general cognition factor, or the *g*-factor, from twelve cognitive scores from different tasks. The *g*-factor underlies variability across cognitive domains and reflects the overall cognition (*Jensen, 2000*; *Panizzon et al., 2014*). Next, we applied machine learning to predict the *g*-factor from 133 mental health indices and 72 neuroimaging phenotypes in unseen participants. For neuroimaging, we created predictive models from both individual neuroimaging phenotypes and phenotypes combined within and across three MRI modalities via stacking. Finally, we conducted commonality analyses (*Nimon et al., 2008*) to quantify the contribution of neural indicators of cognition based on different neuroimaging phenotypes to explaining the relationship between cognition and mental health.

## Results

### *g*-factor-modeling

To model the *g*-factor, we split the data into five outer folds, each comprising training (80% of the data) and test (20% of the data) sets (see *Table 1* for sample characteristics and the Data analysis section).

In each fold, data factorability, assessed using Kaiser-Meyer-Olkin statistics (KMO >0.87) and Bartlett's test of sphericity ($p < 0.05$), indicated good suitability for factor analysis. Parallel analysis suggested that four factors were sufficient to explain the latent structure of the dataset (*Figure 1*). Exploratory structural equation modeling (ESEM) within a confirmatory factor analysis (CFA) further supported the adequacy and construct validity of the resulting factor structure (*Figure 2a*, *Figure 2— figure supplement 1*, *Table 2*).

For each fold, the Comparative Fit Index (CFI) was > 0.96, the Tucker-Lewis Index (TLI) > 0.92, the Root Mean Square Error of Approximation (RMSEA) ≤ 0.05, and the Standardized Root Mean Square Residual (SRMR) < 0.03, indicating good model fit (*Bentler, 1990*; *Hu and Bentler, 1999*; *Xia and Yang, 2019*). Four latent factors captured approximately 27% of the covariance structure of cognitive tests, and the *g*-factor accounted for 39% of the variance in cognitive scores (*Supplementary file 1*).

### Predictive modeling

#### Mental health

On average, information about mental health predicted the *g*-factor at $R^2_{mean} = 0.10$ and $r_{mean} = 0.31$ (95% CI [0.291, 0.315]; *Figure 2b*; *Table 3*).

The magnitude and direction of factor loadings for mental health in the PLSR model allowed us to quantify the contribution of individual mental health indices to cognition. Overall, the scores for mental distress, alcohol and cannabis use, and self-harm behaviours relate positively, and the scores for anxiety, neurological and mental health diagnoses, unusual or psychotic experiences, happiness and subjective wellbeing, and negative traumatic events relate negatively to cognition.

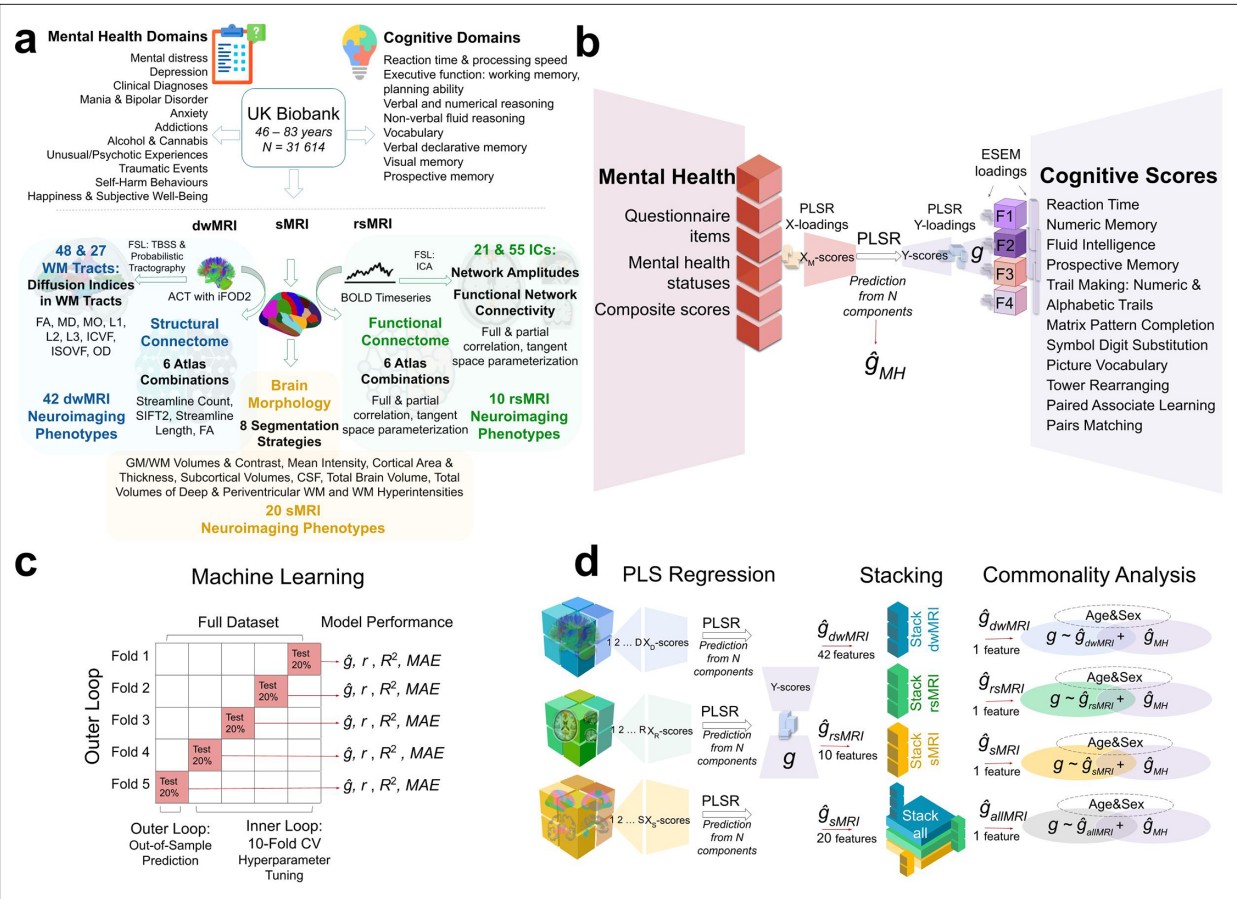

**Figure 1.** Experimental design. (**a**) UK Biobank variables: cognitive tests, mental health, and neuroimaging phenotypes from three Magnetic Resonance Imaging (MRI) modalities. (**b**). Derivation of the *g*-factor from cognitive performance scores with Exploratory Structural Equation Modeling (ESEM) and prediction of the *g*-factor from mental health indices using Partial Least Squares Regression (PLSR). (**c**) Scheme of the machine learning model (PLSR) with nested cross-validation. (**d**) Scheme of the two-level predictive modeling and commonality analyses. First, individual neuroimaging phenotypes from diffusion-weighted MRI (dwMRI) (42 phenotypes), resting-state functional MRI (rsMRI) (10 phenotypes), and structural MRI (sMRI) (20 phenotypes) are used as features to predict the *g*-factor. Then, *g*-factor values predicted from distinct neuroimaging phenotypes are combined within each modality ('dwMRI Stacked,' 'rsMRI Stacked,' and 'Stacked sMRI') as well as across all modalities ('All MRI Stacked') and used as features, resulting in one predicted value per subject per stacked model ($\hat{g}_{dwMRI}$, $\hat{g}_{rsMRI}$, $\hat{g}_{sMRI}$, and $\hat{g}_{allMRI}$). Finally, values predicted from MRI data together with the values predicted from mental health indices ($\hat{g}_{MH}$) are used as independent explanatory variables in commonality analyses. $X_D$-scores, $X_R$-scores, $X_S$-scores, and *Y*-scores, weighted linear combinations of the original features (dwMRI, rsMRI, and sMRI neuroimaging phenotypes, respectively) in PLSR; *WM*, white matter; *TBSS*, tract-based spatial statistics; *ACT*, anatomically-constrained tractography; *iFOD2*, Fiber Orientation Distributions; *FA*, fractional anisotropy; *MD*, mean diffusivity; *MO*, diffusion tensor mode; *L1, L2, L3*, eigenvalues of the diffusion tensor; *ICVF*, intracellular volume fraction; *OD*, orientation dispersion index; *ISOVF*, isotropic volume fraction; *BOLD*, blood oxygenation level dependent signal; *ICA*, independent component analysis; *GM*, grey matter; *CSF*, cerebrospinal fluid; *F1, F2, F3*, and *F4*, latent factors from ESEM; *ESEM loadings*, loadings of the test scores onto the latent factors and loadings of the latent factors onto the *g*-factor; *X-loadings* and *Y-loadings*, loadings of the predictor (mental health measures; X) and target (*g*-factor; Y) variables; respectively, onto the PLSR components; $X_M$-scores and *Y*-scores, the weighted linear combinations of the original predictor (mental health measures) and target (*g*-factor) variables, respectively; $\hat{g}_{MH}$, values of the *g*-factor predicted from mental health features; $\hat{g}$, predicted values of the *g*-factor; *r*, Pearson r (between original and predicted values of the *g*-factor); $R^2$, coefficient of determination (between original and predicted values of the *g*-factor); *MAE*, mean absolute error; *CV*, cross-validation.

## Brain MRI

The predictive performance of neuroimaging phenotypes varied from low to moderate. At the modality level, rsMRI and dwMRI showed the highest and lowest performance, respectively (**Figure 3**). On average, neuroimaging phenotypes stacked within dwMRI, rsMRI, and sMRI predicted the *g*-factor at $R^2_{mean} = 0.073$, 0.105, and 0.095, and $r_{mean} = 0.27$ (95% CI [0.252, 0.273]), 0.33 (95% CI [0.308, 0.331]), and 0.3 (95% CI [0.284, 0.307]), respectively. Stacking all 72 neuroimaging phenotypes boosted the predictive performance of the MRI-based model for cognition, yielding $R^2_{mean} = 0.159$ and $r_{mean} = 0.398$

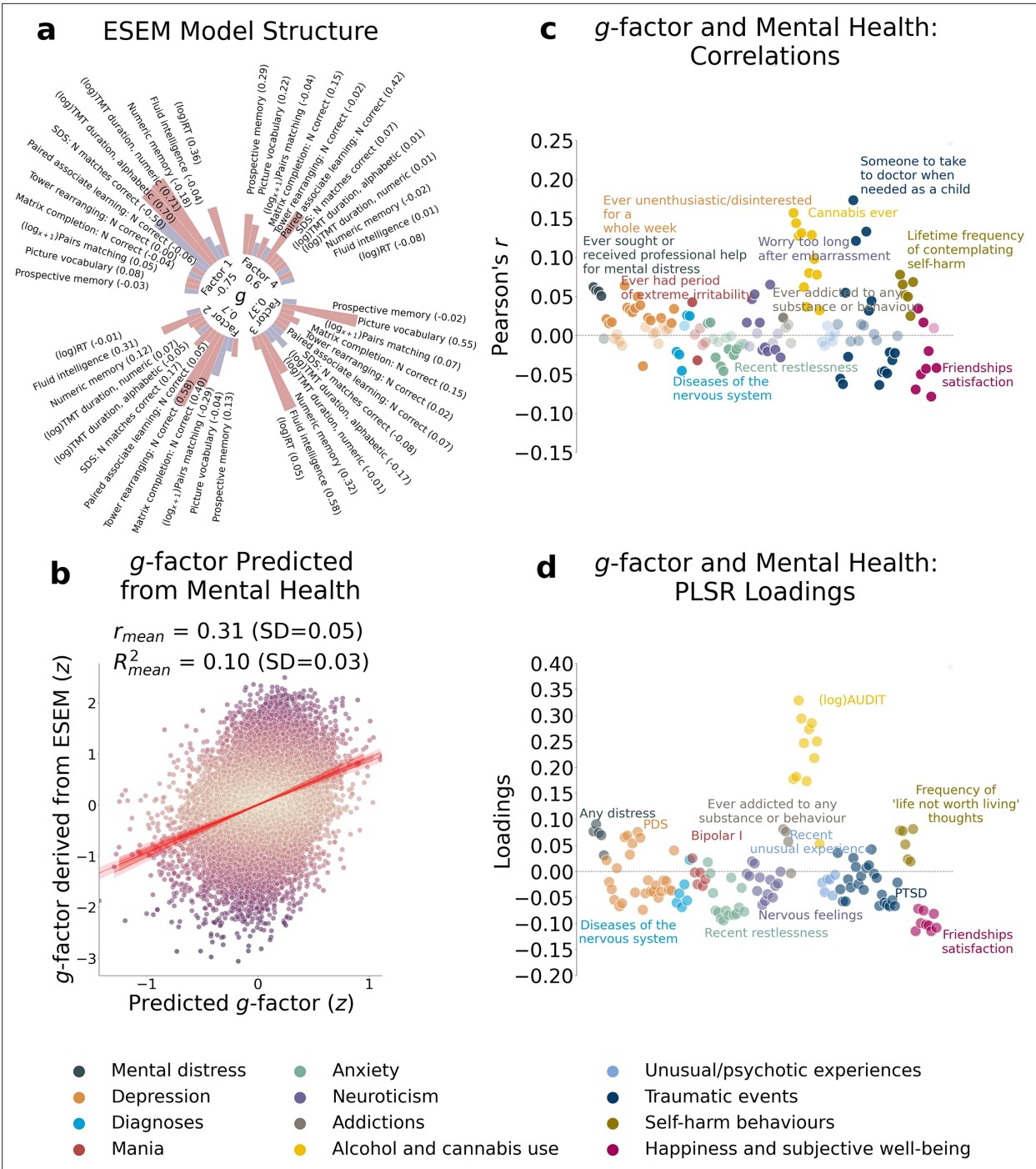

**Figure 2.** *g*-factor modeling and the relationship between the *g*-factor and mental health features. In our analysis, we derived the *g*-factor and built machine learning models in each outer fold separately. For visualization purposes, we display the results of the (**a**) Exploratory Structural Equation Modeling (ESEM) performed on a sample of 31,614 participants and (**b**) Partial Least Squares Regression (PLSR) model for the *g*-factor and mental health built on a sample of 21,077 participants with a single train/test split (80%/20%) as representations of ESEM and PLSR model structures across fivefolds. (**a**) Loadings of the cognitive test scores onto four latent factors and loadings of the latent factors onto the *g*-factor based on the ESEM. (**b**) Scatterplot of the observed *g*-factor and *g*-factor predicted from mental health indices with PLSR. Out-of-sample predictive performance of the PLSR model is evaluated with Pearson $r$ and $R^2$ averaged across fivefolds. (**c**) Pearson correlations between the *g*-factor and mental health indices. Mental health features are grouped into categories. Within each category, a feature with the highest absolute value of Pearson $r$ is annotated. Pale and bright dots represent non-significant and significant correlations, respectively. (**d**) Loadings of mental health indices in the PLSR model showing the relationships between the features (mental health) and the target variable (*g*-factor). The loadings are averaged across all PLSR components and weighted by $R^2$ in

*Figure 2 continued on next page*

*Figure 2 continued*

the training set. Mental health features are grouped into categories. Within each category, a feature with the highest absolute value of the loading is annotated. *SD*, standard deviation (mean across fivefolds).

The online version of this article includes the following source data and figure supplement(s) for figure 2:

**Source data 1.** Source data containing factor loadings, scatterplot data, PLSR results, and the cognitive test correlation matrix underlying all panels of *Figure 2*.

**Figure supplement 1.** Heatmap plot of the correlations between twelve scores from eleven cognitive tests of the UK Biobank cognitive test battery (N=31,614).

(95% CI [0.379, 0.402]; *Figure 4*). The best algorithm for the stacked model was XGBoost. We outline results for each neuroimaging phenotype in *Supplementary file 2* and for each MRI modality and each stacking algorithm in *Table 4*.

## dwMRI

Overall, models based on structural connectivity metrics performed better than TBSS and probabilistic tractography (*Figure 3*). TBSS, in turn, performed better than probabilistic tractography (*Figure 3* and *Supplementary file 2*). The number of streamlines connecting brain areas parcellated with aparc MSA-I had the best predictive performance among all dwMRI neuroimaging phenotypes ($R^2_{mean}$ = 0.052; $r_{mean}$ = 0.227, 95% CI [0.212, 0.235]). To identify features driving predictions, we correlated streamline counts in the aparc MSA-I parcellation with the predicted *g*-factor values from the PLSR model. Positive associations with the predicted *g*-factor were strongest for left superior parietal-left caudal anterior cingulate, left caudate-right amygdala, and left putamen-left hippocampus connections. The most marked negative correlations involved left putamen-right posterior thalamus and right pars opercularis-right caudal anterior cingulate pathways (*Figure 5*, *Figure 5—figure supplement 1*).

The mean length of the streamlines connecting nodes from the Schaefer atlas for 500 cortical areas combined with MSA-IV had the lowest performance among all structural connectivity metrics ($R^2_{mean}$ = 0.018; $r_{mean}$ = 0.145, 95% CI [0.132, 0.156]; *Figure 3*). Among dwMRI IDPs, eigenvalue L2 from TBSS had the best predictive performance ($R^2_{mean}$ = 0.045; $r_{mean}$ = 0.207, 95% CI [0.194, 0.216]), and MO and OD derived with probabilistic tractography were least predictive of the *g*-factor ($R^2_{mean}$ = 0.006; $r_{mean}$ = 0.076, 95% CI [0.065, 0.087] and $R^2_{mean}$ = 0.01; $r_{mean}$ = 0.099, 95% CI [0.085, 0.109], respectively).

Stacking *g*-factor values predicted from all dwMRI neuroimaging phenotypes improved the predictive performance of dwMRI to $R^2_{mean}$ = 0.073 and $r_{mean}$ = 0.265 (95% CI [0.252, 0.273]; *Figure 4* and *Table 4*). The best algorithm for the stacked model was Random Forest.

## rsMRI

Among RSFC metrics for 55 and 21 ICs, tangent parameterization matrices yielded the highest performance in the training set compared to full and partial correlation, as indicated by the cross-validation score. Functional connections between the limbic (IC10) and dorsal attention (IC18) networks, as well as between the ventral attention (IC15) and default mode (IC11) networks, displayed the highest positive association with cognition. In contrast, functional connectivity between the limbic (IC43, the highest activation within network) and default mode (IC11) and limbic (IC45) and frontoparietal (IC40) networks, between the dorsal attention (IC18) and frontoparietal (IC25) networks, and between the

**Table 2.** Goodness-of-fit indices for the hierarchical *g*-factor model across fivefolds.

| Fold | $\chi^2$ | *p*-value for $\chi^2$ | df | CFI | TLI | BIC | RMSEA | SRMR |
|------|----------|------------------------|-----|-------|-------|-------------|--------|-------|
| 1 | 1812.236 | <0.001 | 30 | 0.969 | 0.933 | 805169.905 | 0.048 | 0.026 |
| 2 | 892.577 | <0.001 | 30 | 0.985 | 0.967 | 804456.322 | 0.034 | 0.016 |
| 3 | 805.374 | <0.001 | 30 | 0.987 | 0.971 | 804114.165 | 0.032 | 0.017 |
| 4 | 1335.887 | <0.001 | 30 | 0.978 | 0.951 | 804537.156 | 0.041 | 0.019 |
| 5 | 1926.098 | <0.001 | 30 | 0.967 | 0.928 | 805461.947 | 0.050 | 0.027 |

$\chi^2$, Chi-square test statistic; *df*, degrees of freedom; *CFI*, Comparative Fit Index; *TLI*, Tucker-Lewis Index; *BIC*, Bayesian Information Criteria; *RMSEA*, Root Mean Square Error of Approximation; *SRMR*, Standardised Root Mean Square Residual.

**Table 3.** Out-of-sample predictive performance of mental health features in the Partial Least Squares Regression (PLSR) model across fivefolds.

| | MSE | MAE | $R^2$ | r | p-value |
|---|---|---|---|---|---|
| Fold 1 | 0.425 | 0.521 | 0.142 | 0.377 | <0.01 |
| Fold 2 | 0.621 | 0.624 | 0.061 | 0.25 | <0.01 |
| Fold 3 | 0.696 | 0.667 | 0.059 | 0.244 | <0.01 |
| Fold 4 | 0.448 | 0.531 | 0.12 | 0.347 | <0.01 |
| Fold 5 | 0.448 | 0.527 | 0.114 | 0.34 | <0.01 |
| Mean performance: | 0.53 | 0.57 | 0.10 | 0.31 | |

MSE, mean squared error; MAE, mean absolute error; $R^2$, coefficient of determination; r, Pearson r.

ventral attention (IC15) and frontoparietal (IC40) networks, showed the highest negative association with cognition (**Figure 5**, **Figure 5—figure supplement 2**).

Among RSFC metrics for parcellated time series data, full correlation matrices performed best in the training set. Overall, RSFC between 55 ICs quantified with tangent space parameterization had the highest predictive performance ($R^2_{mean}$ = 0.088, $r_{mean}$ = 0.3, 95% CI [0.284, 0.307]), followed by RSFC between 200 cortical and 16 subcortical regions (Schaefer cortical atlas +MSA I) measured with full correlation ($R^2_{mean}$ = 0.07, $r_{mean}$ = 0.27, 95% CI [0.255, 0.278]). The predictive performance of 21 and 55 ICs amplitudes was the lowest ($R^2_{mean}$ = 0.013, $r_{mean}$ = 0.14, 95% CI [0.098, 0.122] and $R^2_{mean}$ = 0.019, $r_{mean}$ = 0.11, 95% CI [0.125, 0.149], respectively; **Supplementary file 2**).

Stacking g-factor values predicted from rsMRI neuroimaging phenotypes considerably improved the predictive performance of rsMRI to $R^2_{mean}$ = 0.105 and $r_{mean}$ = 0.325 (95% CI [0.308, 0.331]; **Figure 4** and **Table 4**). Similar to dwMRI, the best algorithm in the stacked model was Random Forest.

### sMRI

FreeSurfer subcortical volumetric subsegmentation and ASEG had the highest performance among all sMRI neuroimaging phenotypes ($R^2_{mean}$ = 0.068; $r_{mean}$ = 0.244, 95% CI [0.237, 0.259] and $R^2_{mean}$ = 0.059; $r_{mean}$ = 0.235, 95% CI [0.221, 0.243], respectively). In FreeSurfer subcortical volumetric subsegmentation, volumes of all subcortical structures, except for left and right hippocampal fissures, showed positive associations with cognition. The strongest relations were observed for the volumes of the bilateral whole hippocampal head and whole hippocampus (**Figure 5**, **Figure 5—figure supplement 3**). Gray matter morphological characteristics from ex-vivo Brodmann Area Maps showed the lowest predictive performance ($R^2_{mean}$ = 0.008, $r_{mean}$ = 0.089, 95% CI [0.075, 0.098]; **Figure 3** and **Supplementary file 2**).

Stacking g-factor values predicted from all sMRI neuroimaging phenotypes improved the model's predictive performance to $R^2_{mean}$ = 0.095 and $r_{mean}$ = 0.298 (95% CI [0.284, 0.307]). The best algorithm in the stacked model was SVR (**Figure 4** and **Table 4**).

## Commonality analysis

Different neuroimaging phenotypes captured the relationship between cognition and mental health at varying degrees, as indicated by a percentage ratio between the common effect of mental health-g and neuroimaging-g and the total effect of mental health-g. Neuroimaging phenotypes from dwMRI, rsMRI, and sMRI accounted for 2.1–19.3%, 4–25.8%, and 4.8–21.8% of the cognition-mental health relationship, respectively (**Figure 6**). Among dwMRI neuroimaging phenotypes, the number of streamlines connecting gray matter regions from Destrieux (aparc.a2009s) cortical +MSA I subcortical parcellations shared the highest proportion of variance with mental health-g (19.3%). For rsMRI, the largest proportion of the common effect of mental health-g was shared with RSFC between 55 ICs (25.8%). For sMRI, subcortical volumetric subsegmentation contributed most to the link between cognition and mental health (21.8%). The correlation between the performance of each neuroimaging phenotype in predicting cognition and the proportion of the relationship between cognition and mental health captured by the phenotype was r=0.97 (95% CI [0.958, 0.982]; **Figure 7**).

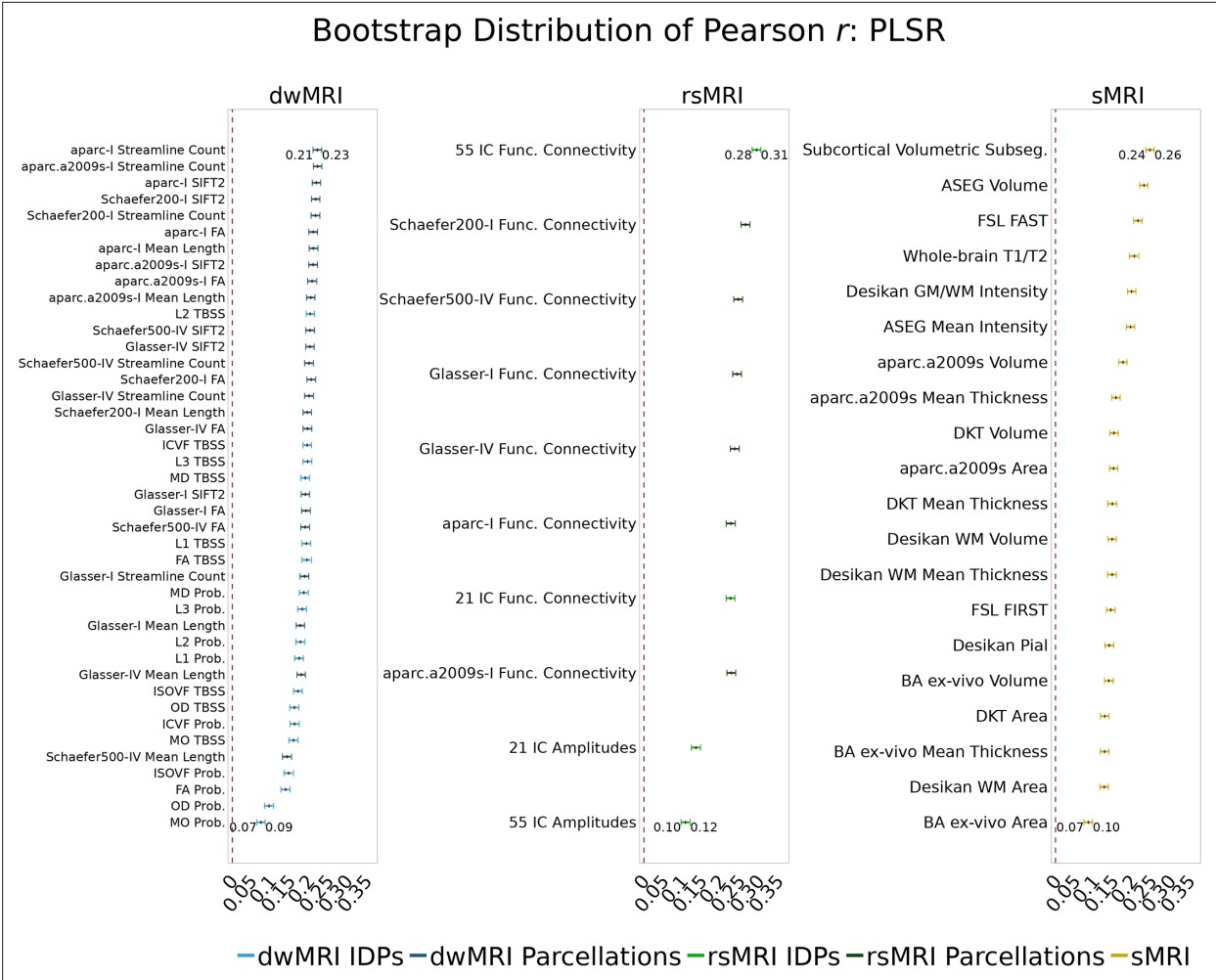

**Figure 3.** Predictive performance of machine learning models based on 72 individual neuroimaging phenotypes. Bootstrap distribution of Pearson *r* for the *g*-factor derived from Exploratory structural equation modeling ESEM and *g*-factor predicted from each neuroimaging phenotype, and corresponding 95% confidence intervals (95% CI). Values at the top and bottom of the plots indicate the lower and upper 95% CI for the bootstrap Pearson *r*.

The online version of this article includes the following source data for figure 3:

**Source data 1.** Source data containing the bootstrapped predictive performance metrics of machine learning models based on dwMRI, rsMRI, and sMRI neuroimaging phenotypes.

When we stacked neuroimaging phenotypes within dwMRI, rsMRI, and sMRI, we captured 25.5%, 29.8%, and 31.6% of the predictive relationship between cognition and mental health, respectively. By stacking all 72 neuroimaging phenotypes across three MRI modalities, we enhanced the explanation to 48% (*Figure 8e–h*).

Age and sex shared substantial overlapping variance with both mental health and neuroimaging in explaining cognition, accounting for 43% of the variance in the cognition-mental health relationship. Multimodal neural marker of cognition based on three MRI modalities ('All MRI Stacked') explained 72% of this age and sex-related variance (*Figure 8i–l*).

## Discussion

Our study is the first to quantify the contribution of the neural indicators of cognition, as reflected by the *g*-factor, to its relationship with mental health in the largest population-level cohort. We show that the performance of each neural indicator in predicting cognition per se is strongly related to its ability to explain the link between cognition and mental health. In other words, the 'robustness' of

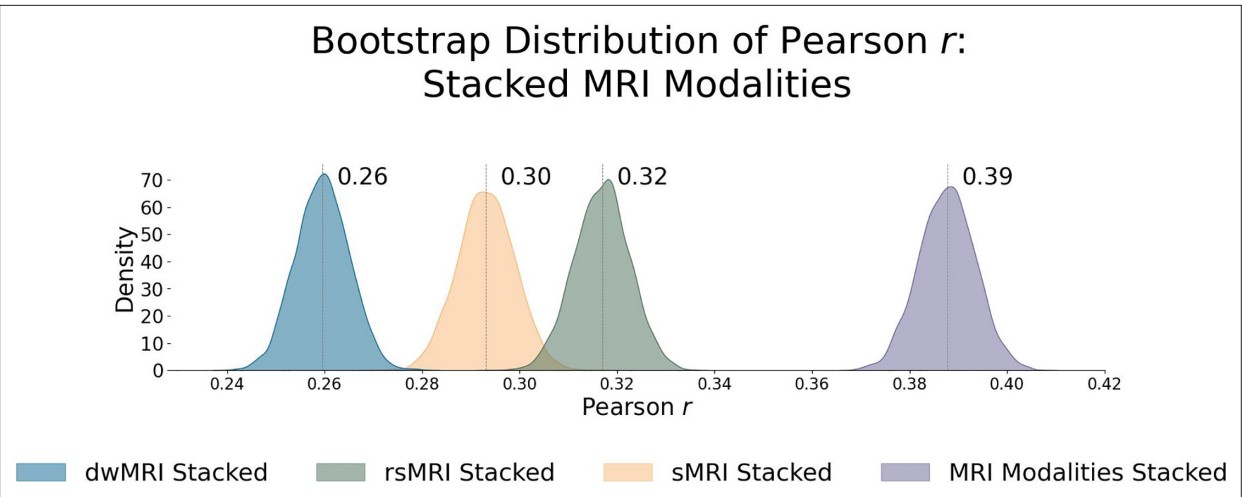

**Figure 4.** Predictive performance of machine learning models based on neuroimaging phenotypes stacked within and across three Magnetic Resonance Imaging (MRI) modalities. Bootstrap distribution of Pearson $r$ between the $g$-factor derived from Exploratory structural equation modeling (ESEM) and the $g$-factor predicted from neuroimaging phenotypes stacked within diffusion-weighted MRI (dwMRI), resting-state functional MRI (rsMRI), structural MRI (sMRI), and across all MRI modalities. Values at the top of each plot mark the median Pearson $r$.

The online version of this article includes the following source data for figure 4:

**Source data 1.** Source data containing the bootstrapped predictive performance metrics of machine learning models based on neuroimaging phenotypes stacked within and across the three MRI modalities.

**Table 4.** Mean (averaged across fivefolds) out-of-sample predictive performance of Magnetic Resonance Imaging (MRI) modalities stacked using four machine learning algorithms.

| | Algorithm | $R^2$ | $r$ | MSE | MAE |
|---|---|---|---|---|---|
| | ElasticNet | 0.027 | 0.227 | 0.97 | 0.782 |
| | **Random Forest** | **0.073** | **0.265** | **0.924** | **0.764** |
| | Support Vector Regression | 0.036 | 0.247 | 0.961 | 0.777 |
| dwMRI | XGBoost | 0.061 | 0.26 | 0.936 | 0.768 |
| | ElasticNet | 0.100 | 0.325 | 0.897 | 0.752 |
| | **Random Forest** | **0.105** | **0.325** | **0.891** | **0.75** |
| | Support Vector Regression | 0.101 | 0.327 | 0.896 | 0.751 |
| rsMRI | XGBoost | 0.102 | 0.326 | 0.895 | 0.751 |
| | ElasticNet | 0.094 | 0.294 | 0.903 | 0.755 |
| | Random Forest | 0.093 | 0.293 | 0.904 | 0.755 |
| | **Support Vector Regression** | **0.095** | **0.298** | **0.902** | **0.753** |
| sMRI | XGBoost | 0.095 | 0.296 | 0.902 | 0.754 |
| | ElasticNet | 0.131 | 0.374 | 0.866 | 0.738 |
| | Random Forest | 0.152 | 0.383 | 0.845 | 0.729 |
| | Support Vector Regression | 0.139 | 0.383 | 0.859 | 0.734 |
| All MRI modalities | **XGBoost** | **0.159** | **0.398** | **0.838** | **0.726** |

$R^2$, coefficient of determination; $r$, Pearson $r$; MSE, mean squared error; MAE, mean absolute error; dwMRI, diffusion-weighted MRI; rsMRI, resting-state MRI; sMRI, T1-weighted and T2-weighted structural MRI. The algorithms that yielded the highest $R^2$ are highlighted in bold.

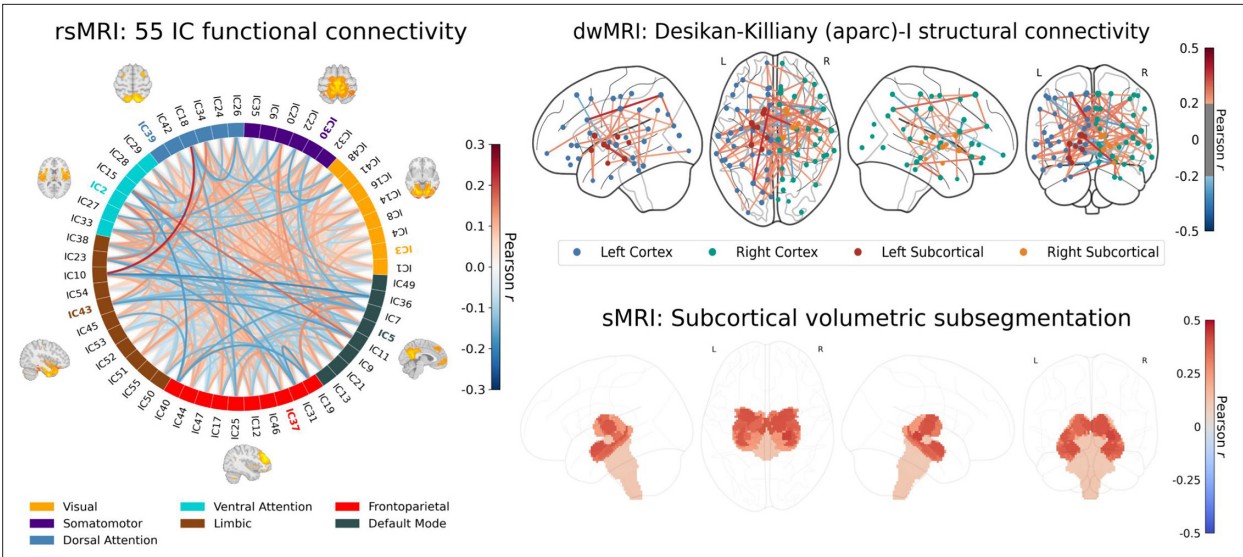

**Figure 5.** Feature importance maps for neuroimaging features with the highest predictive performance for cognition derived via the Haufe transformation. The color of the lines (resting-state functional MRI, rsMRI, and diffusion-weighted MRI, dwMRI) and subcortical structures (sMRI) indicates the magnitude and direction of Pearson correlations between the predicted *g*-factor and features from the top-performing neuroimaging phenotype. Correlations were computed in test sets pooled across five outer folds. **rsMRI:** A connectogram displays network-level feature importance for functional connectivity between 55 neuronally driven independent components (IC) grouped into seven networks (*Thomas Yeo et al., 2011* parcellation). Full correlation matrices rather than tangent space parameterization were used for interpretability. The IC with the highest activation within each network is highlighted in color, and its corresponding functional connectivity map is overlaid. **dwMRI:** The importance of structural connections (streamline count) between brain regions parcellated using the aparc (Desikan-Killiany) MSA-I atlases for predicting cognition is shown as a glass brain plot, with cortical/subcortical nodes (circles) and their connecting edges (lines) colored by correlation direction and strength. **sMRI:** Regional volumes of subcortical structures derived from FreeSurfer subcortical volumetric subsegmentation are overlaid on a glass brain. Values of Pearson *r* for the top correlations are illustrated in *Figure 5—figure supplements 1–3*.

The online version of this article includes the following source data and figure supplement(s) for figure 5:

**Source data 1.** Source data containing feature importance metrics for dwMRI structural connectivity (streamline counts), rsMRI functional connectivity (IC correlations), and sMRI subcortical volumes.

**Figure supplement 1.** Feature importance plot for diffusion-weighted MRI (dwMRI) neuroimaging phenotype with the highest predictive performance for cognition.

**Figure supplement 2.** Feature importance plot for resting-state functional MRI (rsMRI) neuroimaging phenotype with the highest predictive performance for cognition.

**Figure supplement 3.** Feature importance plot for structural MRI (sMRI) neuroimaging phenotype with the highest predictive performance for cognition.

the neural indicator of cognition is associated with its capacity to capture the covariation between cognition and mental health. This means that neural indicators that capture more of the individual differences in cognition also capture more of the cognition–mental health covariation. We further demonstrate that information from 72 neuroimaging phenotypes from three MRI modalities accounts for almost half of the variance in the relationship between cognition and mental health. Aspects of the brain that underpin this relationship are best reflected in neuroimaging phenotypes derived from rsMRI, followed by sMRI and dwMRI. For rsMRI, RSFC between 55 large-scale networks measured with tangent space parameterization was the best-performing neuroimaging phenotype. Among sMRI neuroimaging phenotypes, subcortical grey matter characteristics quantified using FreeSurfer's subcortical volumetric subsegmentation, and ASEG shared the highest proportion of variance that mental health captures. For dwMRI, microstructural properties of the brain's white matter that underlie the link between cognition and mental health were best reflected in the number of streamlines connecting grey matter regions from Destrieux cortical +MSA I subcortical parcellations. Integrating information from neuroimaging phenotypes within and across MRI modalities enhanced both the prediction of cognition and the explanation of the relationship between cognition and mental health. We discuss the results in the context of current knowledge in the field, as follows.

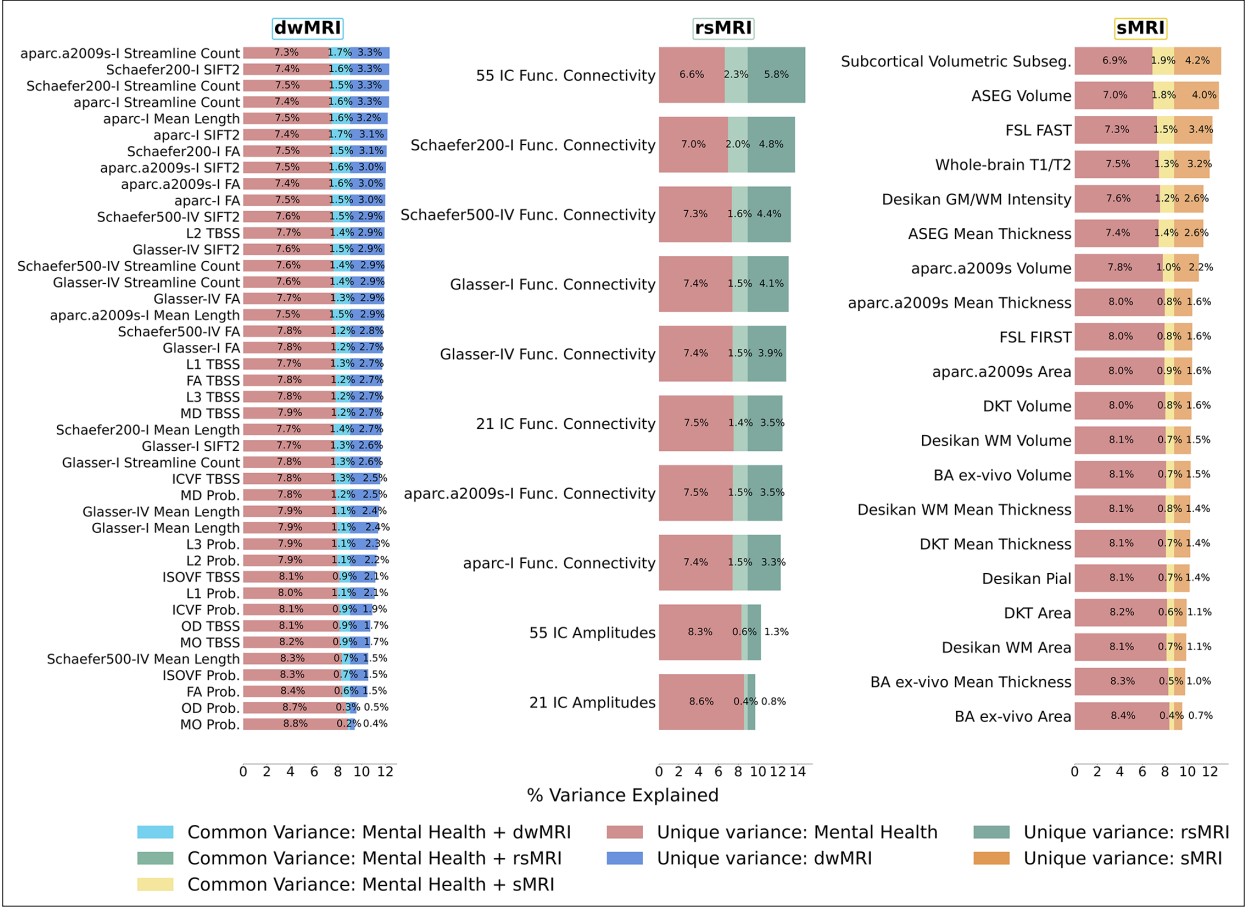

**Figure 6.** Results of commonality analyses: the contribution of neuroimaging phenotypes to the relationship between cognition and mental health. Stacked bar plot diagrams of the results of commonality analyses for each neuroimaging phenotype. *Unique variance*, proportion (%) of variance in the *g*-factor explained uniquely by mental health and neuroimaging phenotypes; *Common variance*, proportion (%) of variance in the *g*-factor shared between mental health and neuroimaging phenotypes; *aparc.a2009s*, Destrieux cortical atlas; *Schaefer7n200p*, Schaefer Atlas for seven networks and 200 parcels; *Schaefer7n500p*, Schaefer Atlas for seven networks and 500 parcels; *I*, Melbourne Subcortical Atlas I; *IV*, Melbourne Subcortical Atlas IV; *SIFT2*, Spherical-Deconvolution Informed Filtering of Tractograms 2; *FA*, fractional anisotropy; *MD*, mean diffusivity; *MO*, diffusion tensor mode; *L1, L2, and L3*, eigenvalues of the diffusion tensor; *OD*, orientation dispersion index; *ICVF*, intracellular volume fraction; *ISOVF*, isotropic volume fraction; *TBSS*, Tract-Based Spatial Statistics; *Func. Connectivity*, functional connectivity; *Subseg.*, FreeSurfer subsegmentation; *ASEG*, FreeSurfer automated subcortical volumetric segmentation; *FSL FAST*, FSL FMRIB's Automated Segmentation Tool; *WM*, white matter; *GM*, gray matter; *FSL, FIRST* FMRIB's Integrated Registration and Segmentation Tool; *DKT*, Desikan-Killiany-Tourville; *BA*, FreeSurfer ex-vivo Brodmann Area Maps.

The online version of this article includes the following source data for figure 6:

**Source data 1.** Source data containing the results of commonality analyses quantifying the contribution of dwMRI, rsMRI, and sMRI neuroimaging phenotypes to the relationship between cognition and mental health.

## The cognition and mental health relationship

Our analysis confirmed the validity of the *g*-factor as a quantitative measure of cognition (***Jensen, 2000***), demonstrating that it captures nearly half (39%) of the variance across twelve cognitive performance scores, consistent with prior studies (***Canivez and Watkins, 2010***; ***Dombrowski et al., 2018***; ***Dubois et al., 2018***; ***Galsworthy et al., 2005***; ***Gignac and Bates, 2017***; ***Gignac and Szodorai, 2024***). Furthermore, we were able to predict cognition from 133 mental health indices, showing a medium-sized relationship that aligns with existing literature (***Abramovitch et al., 2021***; ***Wang et al., 2025***). Although the observed mental health-cognition association is lower than within-sample estimates in conventional regression models, it aligns with our prior mega-analysis in children (***Wang et al., 2025***). Notably, this effect size is not considered small in gerontology. In fact, it falls around the 70th percentile of reported effects and approaches the threshold for a large effect at *r*=0.32 (***Brydges, 2019***). While we focused specifically on cognition as an RDoC core domain, the strength of its relationship with

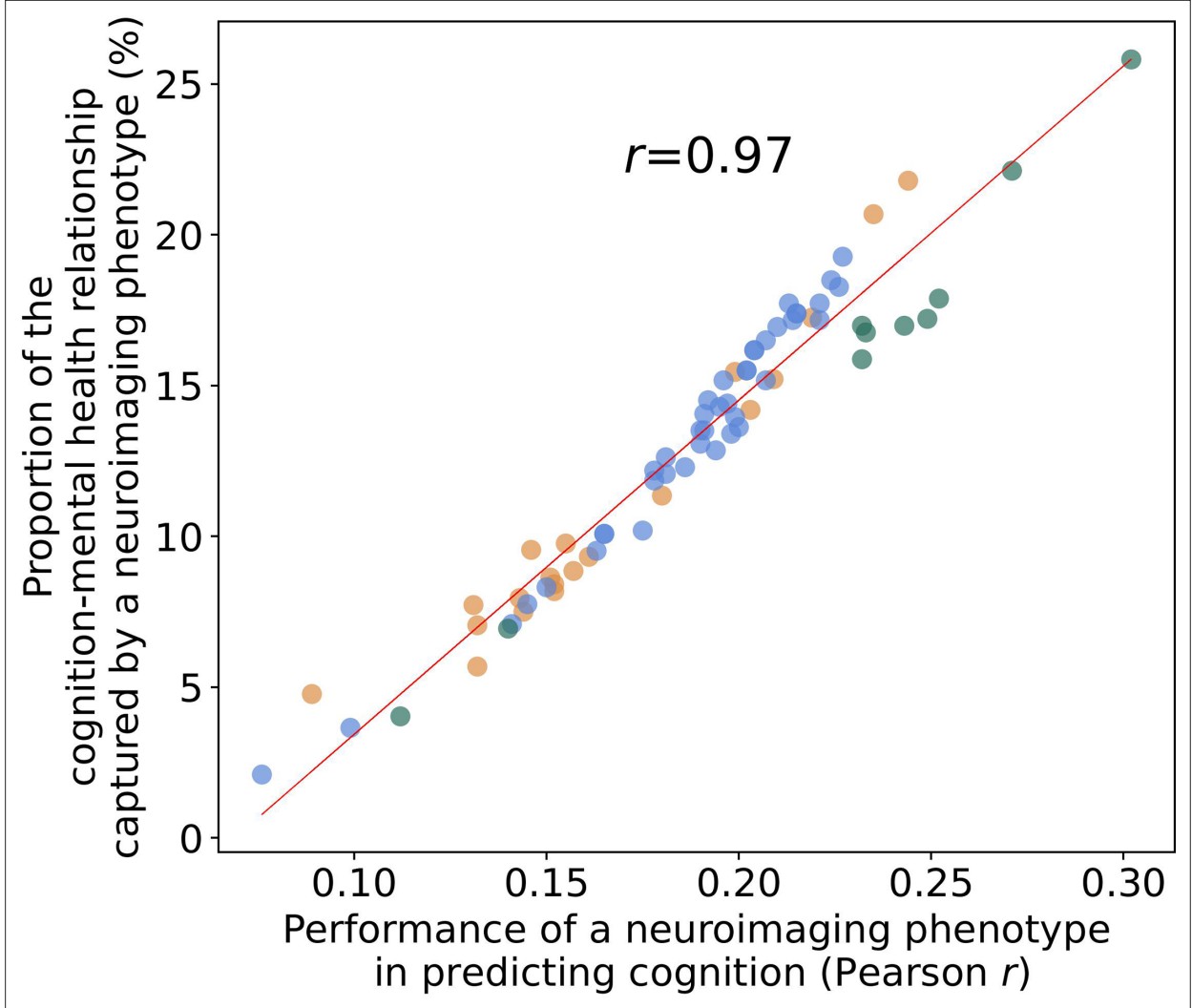

**Figure 7.** Scatterplot of the relationship between the Partial Least Squares Regression (PLSR) performance of individual neuroimaging phenotypes and the proportion of the cognition–mental health relationship they capture.

The online version of this article includes the following source data for figure 7:

**Source data 1.** Source data containing the PLSR performance of individual neuroimaging phenotypes and the proportion of the cognition–mental health relationship captured by these phenotypes.

mental health may be bounded by the influence of other functional domains, particularly in normative, non-clinical samples – a promising direction for future research.

The directions of PLSR loadings were broadly consistent with univariate correlations. PLSR extends beyond univariate approaches by modeling multivariate relationships across features and outcomes. Consistently, both univariate correlations and factor loadings derived from the PLSR model indicated that scores for mental distress, alcohol and cannabis use, and self-harm behaviours related positively, and the scores for anxiety, neurological and mental health diagnoses, unusual or psychotic experiences, happiness and subjective wellbeing, and negative traumatic events related negatively to the $g$-factor. Positive PLSR loadings of features related to mental distress may indicate greater susceptibility to or exaggerated perception of stressful events, psychological overexcitability, and predisposition to rumination in people with higher cognition (*Karpinski et al., 2018*). On the other hand, these findings may be specific to the UK Biobank cohort and the way the questions for this mental health category were constructed. In particular, to evaluate mental distress, the UK Biobank questionnaire asked whether an individual sought or received medical help for or suffered from mental distress. In this regard, the estimate for mental distress may be more indicative of whether an individual experiencing mental distress

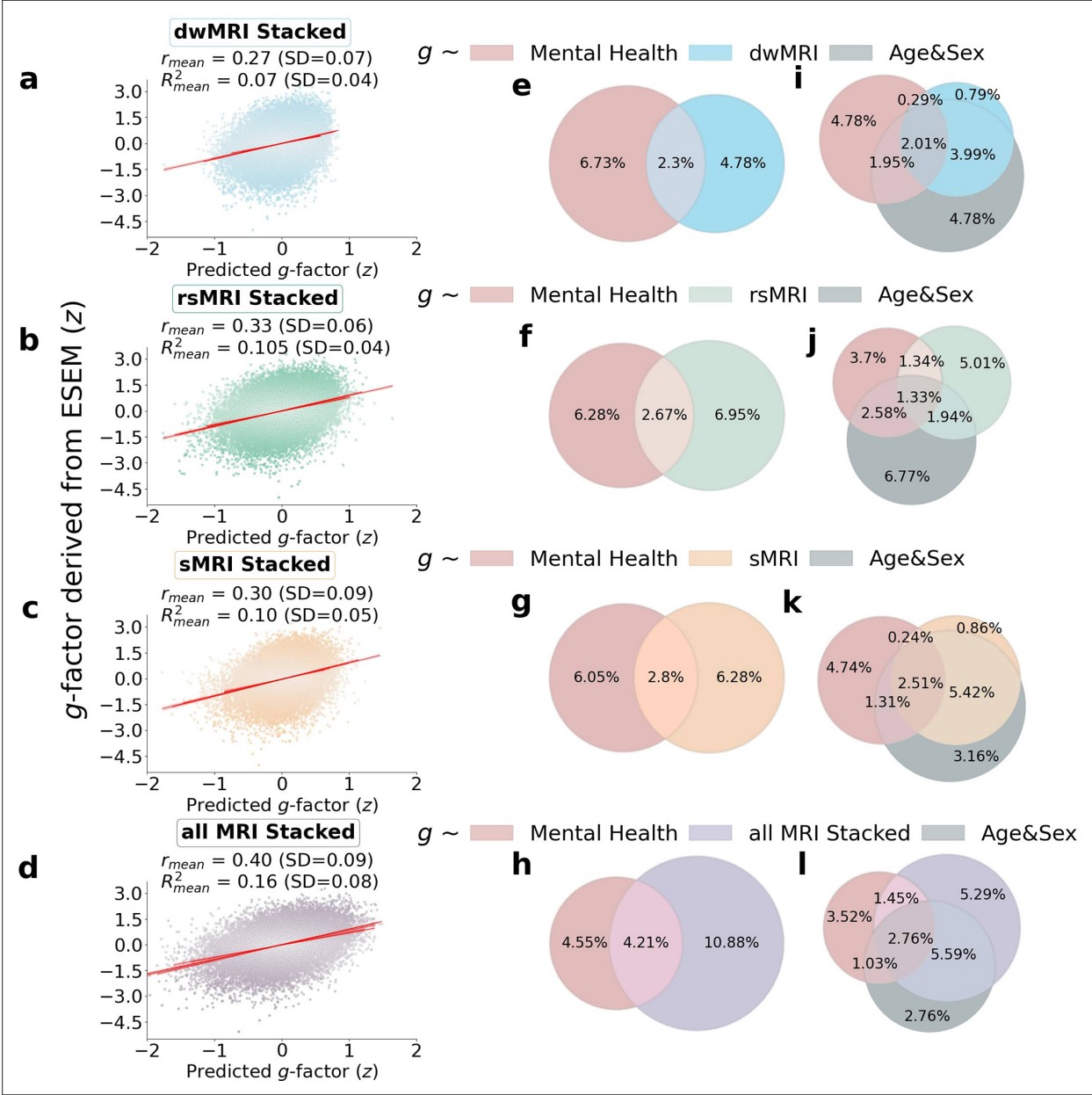

**Figure 8.** The contribution of neuroimaging phenotypes stacked within each and across all Magnetic Resonance Imaging (MRI) modalities to the relationship between cognition and mental health: Results of predictive modeling and commonality analyses. (**a–d**) Distributions of the *g*-factor values derived from cognitive tests via Exploratory structural equation modeling (ESEM) and predicted from neuroimaging phenotypes stacked within diffusion-weighted MRI (dwMRI) (**a**) resting-state functional MRI (rsMRI) (**b**), structural MRI (sMRI) (**c**), and across all MRI modalities (**d**). (**e–l**) Venn diagrams of the results of commonality analyses. (**e–h**) The proportion (%) of variance in the *g*-factor explained uniquely by mental health and neuroimaging phenotypes stacked within dwMRI (**e**), rsMRI (**f**), sMRI (**g**), and across all MRI modalities (**h**), as well as the common effects between mental health and MRI modalities. (**i–l**) The proportion (%) of variance in the *g*-factor explained uniquely by mental health, neuroimaging phenotypes stacked within dwMRI (**i**), rsMRI (**j**), sMRI (**k**), and across all MRI modalities (**l**), and age and sex, as well as the common effects among all explanatory variables.

The online version of this article includes the following source data for figure 8:

**Source data 1.** Source data containing the distributions of the observed and predicted *g*-factors across MRI modalities and the estimates of unique and shared contributions of mental health, neuroimaging phenotypes, and age and sex to the variance in the *g*-factor.

had an opportunity or aspiration to visit a doctor and seek professional help (*Ogueji and Okoloba, 2022*). Thus, people with better cognitive abilities and also with a higher socioeconomic status may indeed be more likely to seek professional help.

Limited evidence supports a positive association between self-harm behaviours and cognitive abilities, with some studies indicating higher cognitive performance as a risk factor for non-suicidal self-harm. Research shows an inverse relationship between cognitive control of emotion and suicidal behaviours that weakens over the life course (*Ogueji and Okoloba, 2022*; *Quintana-Orts et al., 2020*). Some studies have found a positive correlation between cognitive abilities and the risk of non-suicidal self-harm, suicidal thoughts, and suicidal plans that may be independent of or, conversely, affected by socioeconomic status (*Bittár et al., 2020*; *Mars et al., 2014*). In our study, the magnitude of the association between self-harm behaviours and cognition was low (*Figure 2*), indicating a weak relationship.

Positive PLSR loadings of features related to alcohol and cannabis may also indicate the influence of other factors. Overall, this relationship is believed to be largely affected by age, income, education, social status, social equality, social norms, and quality of life (*Cerdá et al., 2011*; *Collins, 2016*; *Druffner et al., 2024*). For example, education level and income correlate with cognitive ability and alcohol consumption (*Druffner et al., 2024*; *Lui et al., 2018*; *Rogne et al., 2021*; *Zhou et al., 2021*). Research also links a higher probability of having tried alcohol or recreational drugs, including cannabis, to a tendency of more intelligent individuals to approach evolutionary novel stimuli (*Kanazawa and Hellberg, 2010*; *Wilmoth, 2012*). This hypothesis is supported by studies showing that cannabis users perform better on some cognitive tasks (*National Academies of Sciences, Engineering, and Medicine, Health and Medicine Division, Board on Population Health and Public Health Practice, & Committee on the Health Effects of Marijuana: An Evidence Review and Research Agenda, 2017*). Alternatively, frequent drinking can indicate higher social engagement, which is positively associated with cognition (*Krueger et al., 2009*). Young adults often drink alcohol as a social ritual in university settings to build connections with peers (*Brown and Murphy, 2020*). In older adults, drinking may accompany friends or family visits (*Beck et al., 2019*; *Kelly et al., 2018*). Mixed evidence on the link between alcohol and drug use and cognition makes it difficult to draw definite conclusions, leaving an open question about the nature of this relationship.

Consistent with previous studies, we showed that anxiety and negative traumatic experiences were inversely associated with cognitive abilities (*Dossi et al., 2020*; *Nyberg et al., 2021*; *Yang et al., 2015*). Anxiety may be linked to poorer cognitive performance via reduced working memory capacity, increased focus on negative thoughts, and attentional bias to threatening stimuli that hinder the allocation of cognitive resources to a current task (*Angelidis et al., 2019*; *Hayes et al., 2012*; *Lukasik et al., 2019*). Individuals with PTSD consistently showed impaired verbal and working memory, visual attention, inhibitory function, task switching, cognitive flexibility, and cognitive control (*Aupperle et al., 2012*; *Johnsen and Asbjørnsen, 2008*; *Khan et al., 2024*; *Moore, 2009*). Exposure to traumatic events that did not reach the PTSD threshold was also linked to impaired cognition. For example, childhood trauma is associated with worse performance in processing speed, attention, and executive function tasks in adulthood, and age at a first traumatic event is predictive of the rate of executive function decline in midlife (*Bomyea et al., 2012*; *Lynch and Lachman, 2020*). In the UK Biobank cohort, adverse life events have been linked to lower cognitive flexibility, partially via depression level (*Petkus et al., 2018*).

In agreement with our findings, cognitive deficits are often found in psychotic disorders (*Künzi et al., 2022*; *McCutcheon et al., 2023*; *Sheffield et al., 2018*). We treated neurological and mental health symptoms as predictor variables and did not stratify or exclude people based on psychiatric status or symptom severity. Since no prior studies have examined isolated psychotic symptoms (e.g. recent unusual experiences, hearing unreal voices, or seeing unreal visions), we avoid speculating on how these symptoms relate to cognition in our sample.

Finally, both negative PLSR loadings and corresponding univariate correlations for features related to happiness and subjective well-being may be specific to the study cohort, as these findings do not agree with some previous research (*Allerhand et al., 2014*; *Jokela, 2022*; *Shi et al., 2022*). On the other hand, our results agree with the study linking excessive optimism or optimistic thinking to lower cognitive performance in memory, verbal fluency, fluid intelligence, and numerical reasoning tasks, and suggesting that pessimism or realism indicates better cognition (*Dawson, 2025*). The concept

of realism/optimism as indicators of cognition is a plausible explanation for a negative association between the *g*-factor and friendship satisfaction, as well as a negative PLSR loading of feelings that life is meaningful, especially in older adults who tend to reflect more on the meaning of life (***Dewitte and Dezutter, 2021***). The latter is supported by the study showing a negative association between cognitive function and the search for the meaning of life and a change in the pattern of this relationship after the age of 60 (***Aftab et al., 2019***). Finally, a UK Biobank study found a positive association of happiness with speed and visuospatial memory but a negative relationship with reasoning ability (***Zhu et al., 2024***).

## How well does brain MRI capture the predictive relationship between cognition and mental health?

Consistent with previous studies, we show that MRI data predict individual differences in cognition with a medium-sized performance ($r \approx 0.4$) (***Dhamala et al., 2021***; ***Gignac and Szodorai, 2024***; ***He et al., 2020***; ***Krämer et al., 2024***; ***Pat et al., 2022***; ***Sripada et al., 2020***; ***Tetereva et al., 2022***). This provides us confidence in using MRI to derive quantitative neuromarkers of cognition. Neural indicators of cognition derived from rsMRI and sMRI neuroimaging phenotypes explain approximately a third of this link (29.8% and 31.6%, respectively), whereas multimodal neural indicators of cognition derived from all neuroimaging phenotypes account for almost half (48%) of the variance in the cognition-mental health relationship. Yet, combining all neuroimaging phenotypes from three MRI modalities allowed us to explain the highest proportion of the variance in cognition that mental health captures.

Among dwMRI-derived neuroimaging phenotypes, models based on structural connectivity between brain areas parcellated with aparc MSA-I (streamline count), particularly connections with bilateral caudal anterior cingulate (left superior parietal-left caudal anterior cingulate, right pars opercularis-right caudal anterior cingulate), left putamen (left putamen-left hippocampus, left putamen-right posterior thalamus), and amygdala (left caudate-right amygdala), result in a neural indicator that best reflects microstructural resources associated with cognition, as indicated by predictive modeling, and more importantly, shares the highest proportion of the variance with mental health-*g*, as indicated by commonality analysis. One of the mechanisms that can be reflected in the link between streamline count and individual variations in cognition is strengthening local connections within a hemisphere and enhancing local (i.e. the connectivity between neighbouring regions) and nodal efficiency (i.e. how well a given brain region is connected to other regions) (***Neudorf et al., 2024***). The somewhat limited utility of diffusion metrics derived specifically from probabilistic tractography in serving as robust quantitative neuromarkers of cognition and its shared variance with mental health may stem from their greater sensitivity and specificity to neuronal integrity and white matter microstructure rather than to dynamic cognitive processes. Critically, probabilistic tractography may be less effective at capturing relationships between white matter microstructure and behavioural scores cross-sectionally, as this method is more sensitive to pathological changes or dynamic microstructural alterations like those occurring during maturation. While these indices can capture abnormal white matter microstructure in clinical populations, such as Alzheimer's disease, schizophrenia, or attention deficit hyperactivity disorder (ADHD) (***Bergamino et al., 2021***; ***Douaud et al., 2011***; ***Silk et al., 2009***), the empirical evidence on their associations with cognitive performance is controversial (***Bozzali et al., 2002***; ***Chen et al., 2023***; ***O'Donnell and Westin, 2011***; ***Patil and Ramakrishnan, 2014***; ***Pierpaoli et al., 1996***; ***Shim et al., 2017***; ***Sripada et al., 2020***; ***Stahl et al., 2007***).

We extend findings on the superior performance of rsMRI in predicting cognition, which aligns with the literature (***Dhamala et al., 2021***; ***Pat et al., 2022***), by showing that it also explains almost a third of the variance in cognition that mental health captures. At the rsMRI neuroimaging phenotype level, this performance is mostly driven by RSFC patterns among 55 ICA-derived networks quantified using tangent space parameterization. At a feature level, these associations are best captured by the strength of functional connections among limbic, dorsal attention and ventral attention, frontoparietal and default mode networks. These functional networks have been consistently linked to cognitive processes in prior research (***Cole et al., 2013***; ***Seeburger et al., 2024***; ***Smallwood et al., 2021***; ***Vossel et al., 2014***).

ICA is a data-driven technique that does not rely on predefined anatomical boundaries. It captures intrinsic large-scale functional networks accounting for individual variability in RSFC. Thus, by providing

more 'personalized' connectivity representations, ICA-derived networks may be more robust neuronal indicators of cognitive performance than node-to-node estimates (*Marrelec and Fransson, 2011*; *Sohn et al., 2015*). Furthermore, using tangent space parametrization to quantify RSFC not only improves the predictive performance of rsMRI for cognition, as shown previously (*Abbas et al., 2023*; *Dadi et al., 2019*; *Ng et al., 2014*; *Simeon et al., 2022*; *Venkatesh et al., 2020*), but also allows us to capture more variance that cognition shares with mental health. Although tangent parametrization matrices do not reflect individual functional brain connections and cannot be interpreted directly, the linearization and projection to Euclidean space make functional connectivity estimates more suitable for statistical analysis and machine learning (*Pervaiz et al., 2020*). Resting-state fluctuation amplitudes from rsMRI, which are the least predictive of cognition and explain the smallest proportion of the variance in cognition that mental health captures, are believed to reflect cardiovascular and cerebrovascular factors distinct from neural effects (*Tsvetanov et al., 2021*). Research indicates that network amplitudes are significantly influenced by age, cardiovascular and lung function, and other physical measures, and are, therefore, subject to high interindividual variability. Consequently, interindividual variability in vascular health and ageing dynamics may confront the capacity of network amplitudes to serve as a robust neural marker of cognition (*Lee et al., 2023*).

Integrating information about brain anatomy by stacking sMRI neuroimaging phenotypes allowed us to explain a third of the link between cognition and mental health. Among all sMRI neuroimaging phenotypes, those that quantified the morphology of subcortical structures, particularly volumes of bilateral hippocampus and hippocampal head, explain the highest portion of the variance in cognition captured by mental health. Our findings show that, at least in older adults, volumetric properties of subcortical structures are not only more predictive of individual variations in cognition but also explain a greater portion of cognitive variance shared with mental health than structural characteristics of more distributed cortical grey and white matter. This aligns with the Scaffolding Theory that proposes stronger compensatory engagement of subcortical structures in cognitive processing in older adults (*Park and Reuter-Lorenz, 2009*; *Reuter-Lorenz and Park, 2014*; *Vieira et al., 2020*).

## Limitations

The study has several limitations. First, the UK Biobank MRI data include only one task for task functional MRI (tfMRI), the Hariri hammer task (*Hariri et al., 2000*), which is not designed to be cognitively demanding. Compared to other MRI modalities, tfMRI from certain tasks, such as the *n*-back working memory task (*Kirchner, 1958*), has been shown to provide the most robust neuromarker of cognition (*Tetereva et al., 2022*). Thus, by not including cognitively demanding tfMRI tasks in predictive models, we may have missed condition-specific variance that may account for a substantial portion of the variance specific to particular cognitive domains. Second, for generalizability purposes, we did not stratify the cohort nor exclude individuals with neurological, cardiovascular, or any other clinically established disorders, assuming that, in older adults, these effects can be tightly intertwined with neuronal mechanisms that sustain cognitive processing. Finally, the UK Biobank's mental health questionnaire and cognitive test battery may miss important information as they cover a limited and specific set of neuropsychiatric conditions and cognitive domains, respectively. For example, the mental health questionnaire does not include questions evaluating autism or ADHD symptoms. The UK Biobank's cognitive test battery was designed specifically for the study and is different from commonly used cognitive test batteries, such as the NIH Toolbox for Assessment of Neurological and Behavioral Function *Gershon et al., 2013* used in the Human Connectome Project (*Elam and Essen, 2022*) and Adolescent Brain Cognitive Development Study (*Casey et al., 2018*) or Wechsler Adult Intelligence Scale *Hartman, 2009* used in the Dunedin Study (*Poulton et al., 2015*), which hinders the cross-study comparison of the cognitive performance score.

## Conclusions

Overall, our findings suggest that RSFC is a more fine-tuned system that exhibits a degree of flexibility and variability that is not entirely constrained by anatomical pathways, as cortical regions can use alternative or parallel pathways to strengthen or weaken functional connections underlying cognitive processing (*Greicius et al., 2009*). Although RSFC is believed to reflect anatomical connectivity (*Greicius et al., 2009*; *O'Reilly et al., 2013*), alterations in structural connectivity do not always compromise RSFC (*O'Reilly et al., 2013*). From this point, a pattern of RSFC maintained for an extended

period may eventually cause concurrent alterations in cognition and mental health, especially if it involves brain areas that have overlapping effects on both. Still, the physical integrity and morphology of the structures providing a 'physical' relay for functionally connected brain areas to communicate play a pivotal role in the brain-behaviour relationship. Although more rigid and less flexible in an adult brain, they determine the amount of neural resources available for cognitive processing. Nevertheless, none of the neuroimaging phenotypes or MRI modalities alone is sufficient to provide a complete picture of the complex relationship between cognition and mental health, and combining all sources of information about brain structure and function reflected in three MRI modalities allows us to derive robust quantitative neuromarkers of cognition and boost the explanation of neural correlates that underlie this link.

In line with the National Institute of Mental Health's RDoC framework, we shed light on the relationship between cognition and mental health as one of the six transdiagnostic spectrums of neuropsychiatric symptoms at one of the seven units of analysis, i.e., the neural level (*Morris and Cuthbert, 2012*). This may serve as a methodological 'bridge' linking other units of analysis, such as genes and behaviour. By elucidating the role of different neuroimaging phenotypes as neural correlates of cognition that overlap with mental health, we provide potential targets for behavioural and physiological interventions that may affect cognition.

Although recent debates *Marek et al., 2022* have challenged the predictive utility of MRI for cognition, our multimodal marker integrating 72 neuroimaging phenotypes captures nearly half of the mental health-explained variance in cognition. We demonstrate that neural markers with greater predictive accuracy for cognition also better explain cognition–mental health covariation, showing that multimodal MRI can capture both a substantial cognitive variance and nearly half of its shared variance with mental health. Finally, we show that our neuromarkers explain a substantial portion of the age- and sex-related variance in the cognition-mental health relationship, highlighting their relevance in modeling cognition across demographic strata.

The remaining unexplained variance in the relationship between cognition and mental health likely stems from multiple sources. One possibility is the absence of certain neuroimaging modalities in the UK Biobank dataset, such as task-based fMRI contrasts, positron emission tomography, arterial spin labelling, and magnetoencephalography/electroencephalography. Prior research has consistently demonstrated strong predictive performance from specific task-based fMRI contrasts, particularly those derived from tasks like the *n*-Back working memory task and the face-name episodic memory task, none of which is available in the UK Biobank (*Kirchner, 1958*; *Pat et al., 2022*; *Rentz et al., 2011*; *Rentz et al., 2011*; *Sripada et al., 2020*; *Tetereva et al., 2022*; *Tetereva et al., 2025*; *Wang et al., 2025*).

Moreover, there are inherent limitations in using MRI as a proxy for brain structure and function. Measurement error and intra-individual variability, such as differences in a cognitive state between cognitive assessments and MRI acquisition, may also contribute to the unexplained variance. According to the RDoC framework, brain circuits represent only one level of neurobiological analysis relevant to cognition (*Insel et al., 2010*). Other levels, including genes, molecules, cells, and physiological processes, may also play a role in the cognition–mental health relationship.

Neuroimaging offers a unique window into the biological mechanisms underlying cognition-mental health overlap – insights unattainable from behavioural data alone. Our findings validate brain-based neural markers as a core unit of analysis for cognitive functioning, advancing mental health research through the lens of cognition. Beyond this conceptual contribution, the study has clinical implications. First, by demonstrating a transdiagnostic link between cognition and mental health, we support interventions that enhance cognition as a pathway to improving mental health. Second, we show neuroimaging as an effective tool for assessing the neurobiological basis of this link. Quantifying neuroimaging's capacity to capture this relationship is essential for future research integrating imaging with cognitive testing to monitor treatment-related neural changes. Such work could enable personalised interventions, using neuroimaging to track cognitive changes and treatment efficacy (e.g. stimulant medications for ADHD) aimed at boosting cognitive functioning.

**Table 5.** Demographics for each subsample analyzed: number, age, and sex of participants who completed all cognitive tests, mental health questionnaires, and Magnetic Resonance Imaging (MRI) scanning.

| | N participants | Age: mean (SD) years | Age: Range | % Females |
|---|---|---|---|---|
| Cognitive Tests | 31 614 | 64.51 (7.66) | 46.0–83.0 | 51.35% |
| Mental Health Questionnaire | 21 077 | 64.63 (7.63) | 47.0–82.0 | 53.0% |
| Cognitive Tests, Mental Health Questionnaire, and dwMRI | 17 250 | 64.25 (7.53) | 47.0–82.0 | 54.68% |
| Cognitive Tests, Mental Health Questionnaire, and rsMRI | 17 005 | 64.2 (7.52) | 47.0–82.0 | 54.92% |
| Cognitive Tests, Mental Health Questionnaire, and sMRI | 14 793 | 64.21 (7.56) | 47.0–82.0 | 54.62% |
| Cognitive Tests, Mental Health Questionnaire, and all MRI | 14 256 | 64.04 (7.49) | 47.0–82.0 | 54.97% |

*SD*, standard deviation.

## Materials and methods

### Data

We used cognition, mental health, and MRI data collected at the first imaging visit (2014–2019) and additional mental health data collected online from the UK Biobank (UK Biobank Resource Application 70132), a prospective epidemiological study involving individuals aged 40–69 years at recruitment. All UK Biobank participants provided informed consent directly to UK Biobank at recruitment. UK Biobank has ethical approval from the North West Multi-centre Research Ethics Committee (reference 16/NW/0274) as a Research Tissue Bank. The analyses included participants who had all brain MRI scans, performed all cognitive tests, and completed mental health questionnaires (*Table 5*).

### Cognition
#### Core measures

To derive the *g*-factor, we used twelve scores from eleven cognitive tests representing the following cognitive domains: reaction time and processing speed, working memory, verbal and numerical reasoning, executive function, non-verbal fluid reasoning, processing speed, vocabulary, planning

**Table 6.** Cognitive tests and core measures of the UK Biobank cognitive test battery used in the study.

| Test | Cognitive domain | Core measures | Field ID |
|---|---|---|---|
| Reaction Time | Reaction time and processing speed | Mean time to correctly identify matches | 20023 |
| Numeric Memory | Working memory | Maximum digits remembered correctly | 4282 |
| Fluid Intelligence | Verbal and numerical reasoning | Fluid intelligence score | 20016 |
| Prospective Memory | Prospective memory | Initial answer | 4292 |
| Trail Making | Executive function | Duration to complete numeric path (trail 1)<br>Duration to complete alphabetic path (trail 2) | 6348<br>6350 |
| Matrix Pattern Completion | Non-verbal fluid reasoning | Number of puzzles correctly solved | 6373 |
| Symbol Digit Substitution | Processing speed | Number of symbol digit matches made correctly | 23324 |
| Picture Vocabulary | Vocabulary (crystallized cognitive ability) | Specific cognitive ability | 26302 |
| Tower Rearranging | Planning abilities (a component of executive function) | Number of puzzles correct | 21004 |
| Paired Associate Learning | Verbal declarative memory | Number of word pairs correctly associated | 20197 |
| Pairs Matching | Visual memory | Number of incorrect matches in round | 399 |

abilities, verbal declarative memory, prospective memory, and visual memory (*Figure 1*, *Table 6*; *Fawns-Ritchie and Deary, 2020*).

## Transformations of cognitive scores

To reduce skewness, we applied log(x) transformations to the mean time required to correctly identify matches in the Reaction Time test and to the time required to complete numeric and alphabetic trails in the Trail Making test (*Fawns-Ritchie and Deary, 2020*; *Lyall et al., 2016*; *Williams et al., 2023*). For the Pairs Matching task, we selected the number of incorrect matches in the six-pair version of the test as the outcome measure and applied a log(x+1) transformation due to the greater variability in this version and the very low rate of incorrect responses in the three-pair condition (*Fawns-Ritchie and Deary, 2020*). In the Prospective Memory test, participants were scored as 0 if they failed to complete the task on the first attempt (i.e. selected a shape other than the target orange circle) and as 1 if they correctly selected the target shape on the initial attempt. Accordingly, all non-target shapes (0 – blue square, 1 – pink star, 2 – gray cross) were recoded as 0, while the target shape (3 – orange circle) was recoded as 1 (*Table 7*).

Negative values were replaced with NA, as they indicated invalid or abandoned trials. In addition, zero values were replaced with NA in tests where a zero score reflected an invalid trial rather than a true performance score (e.g. Fluid Intelligence score, specific cognitive ability in the Picture Vocabulary test, and trail completion time in the Trail Making test).

## g-factor modeling

To capture overall variability in cognition, we built a hierarchical *g*-factor model using exploratory structural equation modeling (ESEM). In this model, the *g*-factor underlies a set of latent factors which, in turn, underlie 12 cognitive scores (*Figures 1 and 2*). Analysis was conducted in *R* (version 4.4.2) within *RStudio* (version 2022.02.0).

To examine data factorability (i.e. suitability for factor analysis), we calculated the Kaiser-Meyer-Olkin (KMO) measure of sampling adequacy and performed Bartlett's test of sphericity (*Bartlett, 1954*; *Kaiser, 1974*). We then conducted exploratory factor analysis (EFA) to determine the number of latent factors and the loadings of cognitive test scores onto these factors. The number of factors was identified using parallel analysis, which compares eigenvalues of the observed correlation matrix with those obtained from randomly generated matrices of the same size (*fa.parallel* function of the *psych* package) (*Horn, 1965*).

The EFA solution was subsequently evaluated using exploratory structural equation modeling within confirmatory factor analysis (ESEM-within-CFA; *esem_efa* function of the *esemComp* package), and anchor variables obtained at this stage were used to construct and fit a hierarchical model. Factor loadings were retained and refined using CFA with the maximum likelihood (ML) estimator and the NLMINB optimization algorithm (Nonlinear Minimization using the Broyden-Fletcher-Goldfarb-Shanno method). Model fit was assessed using the Comparative Fit Index (CFI), Tucker-Lewis Index (TLI), Root Mean Square Error of Approximation (RMSEA), Bayesian Information Criterion (BIC), Standardized Root Mean Square Residual (SRMR), and chi-square ($\chi^2$) statistics. The *g*-factor scores were obtained using the *lavaan* package (*Jackson et al., 2009*; *Pavlov et al., 2021*; *Penny et al., 2007*; *Rosseel, 2012*; *Schwarz, 1978*; *Xia and Yang, 2019*).

The chi-square test evaluates discrepancies between the population covariance matrix and the model-implied covariance matrix. Large $\chi^2$ values relative to the degrees of freedom indicate substantial differences between the empirical and model-implied covariance matrices, while a significant *p*-value leads to rejection of the null hypothesis that these differences are zero. However, because the $\chi^2$ statistic is sensitive to sample size, significant results may occur even when model fit is acceptable in large samples (*Tucker and Lewis, 1973*).

Fit indices compare the target model with a baseline (independence) model that assumes no correlations among observed variables. The TLI, also known as the non-normed fit index, evaluates relative model fit by comparing $\chi^2$ values of the baseline and target models while accounting for model complexity. TLI values above 0.90 indicate good model fit (*Peugh and Feldon, 2020*). The CFI, an adjusted version of the Relative Noncentrality Index, corrects for underestimation of fit in small samples; values above 0.97 indicate good fit and values above 0.95 indicate acceptable fit, although some studies recommend 0.95 as a general threshold (*Peugh and Feldon, 2020*; *Xia and Yang, 2019*).

**Table 7.** Whole-sample distributions of cognitive performance scores used to derive the *g*-factor (N=31,614).

| No | Variable | Statistics/Values | Frequencies | Distribution plot |
|----|----------|-------------------|-------------|-------------------|
| 1 | Reaction time (log(x)) | Mean (SD): 6.4 (0.2) min<med<max: 5.8<6.4<7.4 IQR (CV): 0.2 (0) | 550 distinct values | |
| 2 | Fluid intelligence score | Mean (SD): 6.6 (2) min <med<max: 1<7<13 IQR (CV): 3 (0.3) | 13 distinct values | |
| 3 | Numeric memory: Maximum digits remembered correctly | Mean (SD): 6.8 (1.3) min<med<max: 2<7<12 IQR (CV): 2 (0.2) | 11 distinct values | |
| 4 | Trail making: Duration to complete numeric path (log(x)) | Mean (SD): 5.4 (0.3) min<med<max: 4.5<5.3<7.5 IQR (CV): 0.3 (0.1) | 638 distinct values | |
| 5 | Trail making: Duration to complete alphabetic path (log(x)) | Mean (SD): 6.3 (0.4) min<med<max: 5.1<6.2<8.7 IQR (CV): 0.5 (0.1) | 1542 distinct values | |
| 6 | Symbol digit substitution: Number of correct matches | Mean (SD): 18.9 (5.2) min<med<max: 0<19<37 IQR (CV): 7 (0.3) | 38 distinct values | |
| 7 | Paired associate learning: Number of correct pairs | Mean (SD): 7 (2.5) min<med<max: 0<7<10 IQR (CV): 4 (0.4) | 11 distinct values | |
| 8 | Tower rearranging: Number of puzzles correct | Mean (SD): 9.9 (3.2) min<med<max: 0<10<18 IQR (CV): 4 (0.3) | 19 distinct values | |
| 9 | Matrix pattern completion: Number of puzzles correct | Mean (SD): 8 (2.1) min<med<max: 0<8<15 IQR (CV): 2 (0.3) | 16 distinct values | |

*Table 7 continued on next page*

*Table 7 continued*

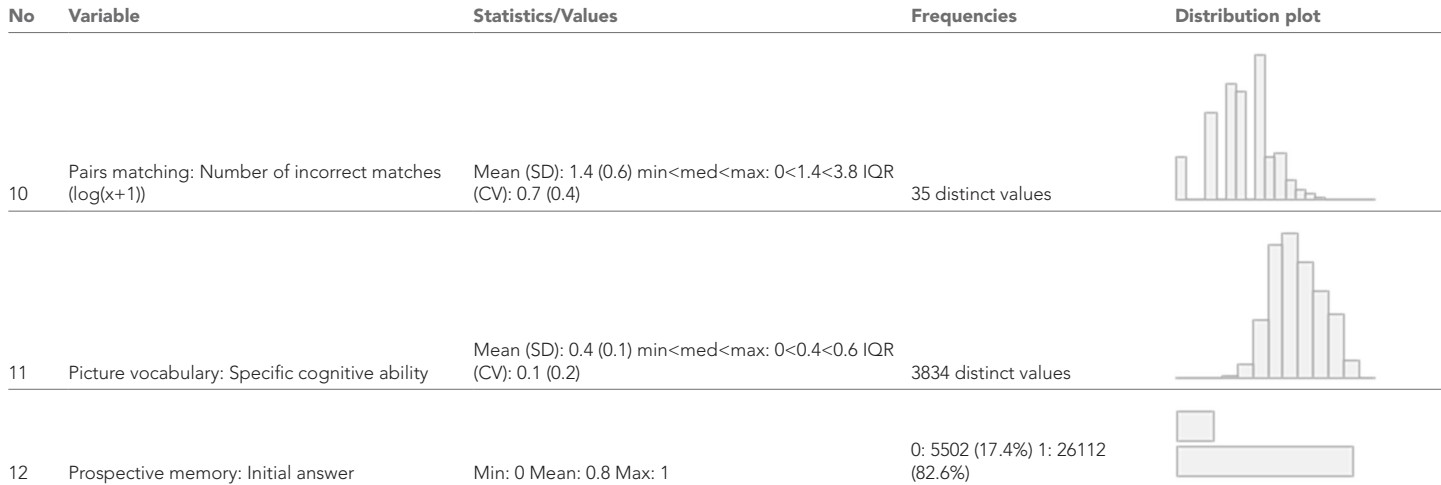

| No | Variable | Statistics/Values | Frequencies | Distribution plot |
|---|---|---|---|---|
| 10 | Pairs matching: Number of incorrect matches (log(x+1)) | Mean (SD): 1.4 (0.6) min<med<max: 0<1.4<3.8 IQR (CV): 0.7 (0.4) | 35 distinct values | |
| 11 | Picture vocabulary: Specific cognitive ability | Mean (SD): 0.4 (0.1) min<med<max: 0<0.4<0.6 IQR (CV): 0.1 (0.2) | 3834 distinct values | |
| 12 | Prospective memory: Initial answer | Min: 0 Mean: 0.8 Max: 1 | 0: 5502 (17.4%) 1: 26112 (82.6%) | |

*SD*, standard deviation; *IQR*, interquartile range; *CV*, coefficient of variation.

The BIC was used for model comparison, with consistent BIC values across fivefolds indicating stable model performance. The BIC penalises model complexity by favouring models that achieve a good fit with fewer parameters, making it well-suited for comparing hierarchical factor models (*Spiegelhalter et al., 2002*). The RMSEA assesses approximate model fit per degree of freedom, with values ≤0.05 indicating close fit, 0.05–0.08 acceptable fit, 0.08–0.10 mediocre fit, and >0.10 poor fit. The SRMR measures the average standardized difference between observed and model-implied correlations and is less sensitive to variable scaling than the Root Mean Square Residual; values <0.05 indicate good fit and values <0.10 acceptable fit (*Pavlov et al., 2021*; *Schermelleh-Engel et al., 2003*).

## Mental health

Mental health measures encompassed 133 variables from twelve groups: mental distress, depression, clinical diagnoses related to the nervous system and mental health, mania (including bipolar disorder), neuroticism, anxiety, addictions, alcohol and cannabis use, unusual/psychotic experiences, traumatic events, self-harm behaviors, and happiness and subjective wellbeing (*Figure 1*, *Supplementary file 3*, and *Appendix 1—table 1*). We included both self-report questionnaire items from all participants and composite diagnostic scores computed following *Davis et al., 2020* and *Dutt et al., 2022* as features in our first-level PLSR model (for explanation, see Data Analysis section). This approach leverages PLSR's ability to handle multicollinearity through dimensionality reduction, enabling the simultaneous use of granular symptom-level information and robust composite. We assess the contribution of each mental health index to general cognition by examining the direction and magnitude of its PLSR-derived loadings on the identified latent variables.

## Description of composite measures

Composite diagnostic scores included the Generalized Anxiety Disorder scale (GAD-7), the Posttraumatic Stress Disorder Checklist (PCL-6), the Alcohol Use Disorders Identification Test (AUDIT), the Patient Health Questionnaire (PHQ-9) (*Davis et al., 2020*), Eysenck Neuroticism (N-12), Probable Depression Status (PDS), and Recent Depressive Symptoms (RDS-4) scores (*Dutt et al., 2022*; *Smith et al., 2013a*). GAD-7, PCL-6, AUDIT, and PHQ-9 scores were derived from items collected during the online follow-up assessment (*Davis et al., 2020*), whereas N-12, PDS, and RDS-4 scores were calculated using baseline assessment data (*Dutt et al., 2022*; *Smith et al., 2013b*).

Depression and GAD variables were further categorized according to frequency, current status (lifetime and current presence of depression or anxiety), severity, and clinical diagnosis confirmed by a healthcare practitioner. Additional depression-related variables distinguished between different subtypes, such as recurrent depression and depression triggered by bereavement or loss. Self-harm

variables were stratified based on lifetime history of self-harm with or without intent to die (*Davis et al., 2020*).

To improve the interpretability of response scales, well-being domain responses were coded so that lower scores indicated lower levels of satisfaction (e.g. 'Extremely unhappy') and higher scores reflected higher satisfaction (e.g. 'Extremely happy'). For all variables, responses 'Prefer not to answer' (−818 for in-person assessment and −3 for online questionnaire) and 'Do not know' (−121 for in-person assessment and −1 for online questionnaire) were replaced with median values. The 'Work/ job satisfaction' variable was excluded from mental health derivatives because the response option 'Not employed' could not be meaningfully coded.

PTSD risk was estimated using PCL-6 questionnaire items. PCL-6 scores ranged from 6 to 29, with scores ≤12 indicating low risk of meeting Clinician-Administered PTSD Scale diagnostic criteria, 13–16 indicating increased risk, 17–25 indicating high risk, and >26 indicating very high risk (*Davis et al., 2020*; *Weathers et al., 2018*). PTSD status was coded as positive for PCL-6 scores ≥14 and included stressful life events in addition to catastrophic trauma.

Alcohol use, dependence, and alcohol-related harm were assessed using the AUDIT questionnaire, calculated as the sum of ten items (*Saunders et al., 1993*). The AUDIT score was further partitioned into alcohol consumption (AUDIT-C; items 1–3) and alcohol-related problems (AUDIT-P; items 4–10) (*Sanchez-Roige et al., 2019*). For items 2–10, missing responses were replaced with zero when the first screening question ('Frequency of drinking alcohol') indicated no alcohol consumption, as these responses correspond to 'Never.' An AUDIT score of 8 was used as a threshold for hazardous alcohol use, with scores of 8–15 indicating harmful drinking and scores >15 indicating alcohol dependence or moderate-to-severe alcohol use disorder (*Babor et al., 2001*; *Saunders et al., 1993*). Hazardous alcohol use and alcohol dependence were, therefore, defined as AUDIT scores ≥8 and ≥15, respectively. 'Alcohol dependence ever' status was coded as positive if a participant reported a history of physical dependence on alcohol. To reduce skewness, AUDIT, AUDIT-C, and AUDIT-P scores were log(x+1)-transformed (*Sanchez-Roige et al., 2019*).

## Brain MRI

MRI data were collected on a Siemens Magnetom Skyra 3T scanner with the Siemens 32-channel head coil across four imaging centres: Cheadle, Reading, Newcastle, and Bristol. The data were processed in FSL (FMRIB Software Library).

For MRI-based models, we used dwMRI, rsMRI, and sMRI (*Figure 1* and *Supplementary file 4*). MRI acquisition protocols are available at http://biobank.ctsu.ox.ac.uk/crystal/refer.cgi?id=2367. MRI processing pipelines are described in the UK Biobank brain imaging documentation (https://biobank.ctsu.ox.ac.uk/crystal/crystal/docs/brain_mri.pdf) and discussed elsewhere (*Alfaro-Almagro et al., 2018*; *Mansour et al., 2023*). MRI data processed by the UK Biobank are available as imaging-derived phenotypes (IDPs). In addition to the UK Biobank's IDPs, we used structural connectomes and BOLD time series created by Mansour and colleagues (*Mansour et al., 2023*).

### Diffusion-weighted MRI (dwMRI)

dwMRI data were acquired using a spin-echo echo-planar imaging sequence with 3x multislice (multiband) acceleration in 100 diffusion-encoding directions across two shells: 50 directions with $b$=1000 s/mm² and 50 directions with $b$=2000 s/mm² (TE = 92 ms, TR = 3600 ms, 2 mm isotropic voxels, partial Fourier 6/8, FOV: 104×104×72, $\delta$=21.4 ms) in two phase-encoding directions. Primary data were collected with anterior-to-posterior phase encoding, and three $b$=0 images with reversed phase encoding were acquired for field map estimation and distortion correction.

To quantify dwMRI, we used 42 IDPs. These IDPs were obtained from head motion-, eddy current-, and gradient distortion-corrected dwMRI data and included fractional anisotropy (FA), diffusion tensor mode (MO), mean diffusivity (MD), and eigenvalues of the diffusion tensor (L1, L2, L3) derived from diffusion tensor fitting, as well as intracellular volume fraction (ICVF), isotropic/free water volume fraction (ISOVF), and orientation dispersion index (OD) from neurite orientation dispersion and density imaging (NODDI). Diffusion metrics were computed for 48 and 27 white matter tracts reconstructed using tract-based spatial statistics (TBSS) and probabilistic tractography, respectively (*Andersson and Sotiropoulos, 2016*; *Behrens et al., 2003*; *Behrens et al., 2007*; *Glasser et al., 2013*; *Jbabdi et al., 2012*; *Smith et al., 2006*).

FA values range from 0 to 1 and reflect the degree of directional water diffusion (anisotropy). MD represents the directionally averaged diffusion rate, independent of orientation (*Alexander et al., 2007*; *O'Donnell and Westin, 2011*; *Soares et al., 2013*). MO is perpendicular to FA. It quantifies tensor shape and distinguishes between linear (tube-like) and planar (disk-like) diffusion patterns (*Tae et al., 2018*; *Yoncheva et al., 2016*). L1, L2, and L3 represent diffusion along the three principal axes of the diffusion tensor (*O'Donnell and Westin, 2011*). NODDI-derived ICVF measures water diffusion within neurites, reflecting neurite density; ISOVF represents freely diffusing water (e.g. cerebrospinal fluid); and OD measures neurite dispersion (spatial configuration), with higher values typically observed in gray matter and lower values in aligned white matter tracts (*Colgan et al., 2016*; *Deligianni et al., 2016*; *Zhang et al., 2012*).

Probabilistic tractography was performed in FSL using BEDPOSTx and PROBTRACKx, and diffusion metrics were mapped onto 27 fibre tracts using standard-space region-of-interest masks defined by AutoPtx (*de Groot et al., 2013*). In probabilistic tractography, white matter tracts are reconstructed by estimating probability distributions of voxel-to-voxel connections to account for uncertainty in fibre orientation (*Morris et al., 2008*). FA images derived from diffusion tensor fitting were also processed using TBSS (*Smith et al., 2006*). In TBSS, individual FA images are projected onto a standard-space mean FA skeleton representing fibre tracts common across participants, and the resulting spatial transformation is applied to other microstructural indices to enable between-subject comparisons (*Smith et al., 2006*). Skeletonized FA maps and DTI/NODDI metrics were averaged across 48 standard-space tract masks from the Susumu Mori white matter atlas (*Mori et al., 2006*; *Wakana et al., 2007*).

In addition to tract-based metrics, we used structural connectomes derived from six combinations of cortical and subcortical atlases: Desikan-Killiany (aparc)+Melbourne Subcortical Atlas scale I (MSA-I), Destrieux (aparc.a2009s)+MSA I, Glasser +MSA I, Glasser+MSA IV, Schaefer-200 (7 networks)+MSA I, and Schaefer-500 (7 networks)+MSA IV (*Desikan et al., 2006*; *Destrieux et al., 2010*; *Glasser et al., 2016*; *Schaefer et al., 2018*; *Thomas Yeo et al., 2011*; *Tian et al., 2020*). Structural connectome data consisted of connectivity matrices for each atlas combination containing streamline count, fibre bundle capacity (SIFT2), mean streamline length, and mean FA for each pair of nodes (*Supplementary file 4*; *Mansour et al., 2023*).

A procedure for computing structural and functional connectomes is discussed in detail in *Mansour et al., 2023*. Briefly, the connectomes were generated following the Anatomically Constrained Tractography (ACT) framework using fibre orientation distributions (iFOD2) probabilistic fibre tracking. In this approach, streamlines are propagated from seed points within white matter or at the gray-white matter interface based on local fibre orientation distributions or dynamic seeding using spherical-deconvolution informed filtering of tractograms (SIFT/SIFT2) (*Smith et al., 2012*; *Smith et al., 2013a*; *Smith et al., 2015*). SIFT and SIFT2 correct for biases in streamline density by aligning tractogram density with the underlying fibre density estimated from the diffusion signal, thereby improving the biological accuracy of reconstructed pathways (*Smith et al., 2013b*; *Smith et al., 2015*). SIFT2 assigns weights to streamlines, and the sum of these weights represents fibre bundle capacity, corresponding to the estimated bundle's total intracellular cross-sectional area and connectivity strength of white matter pathways (*Smith et al., 2020*).

In preprocessing, averaged *b*=0 volumes were skull-stripped using FSL BET to improve T1-weighted and dwMRI registration. Tissue response functions for white matter, gray matter, and cerebrospinal fluid were used to estimate fibre orientation distributions and perform intensity normalization and bias field correction. Whole-brain probabilistic tractography was then conducted within the ACT framework using tissue-type segmentation obtained from FreeSurfer and FSL FIRST, with streamline seeding initiated at the gray-white matter interface, resulting in 10 million streamlines satisfying ACT constraints (*Mansour et al., 2023*; *Smith et al., 2012*).

Whole-brain tractograms were subsequently parcellated using cortical and subcortical atlases defined in template space and transformed into native space. Subcortical parcellations were generated by applying the inverse native T1w-to-MNI152 warp using FSL *applywarp*, while cortical labels were aligned to individual cortical surfaces using FreeSurfer *fsnative* (*Mansour et al., 2023*). Cortical and subcortical parcellations were combined to construct structural connectomes, enabling calculation of streamline-based connectivity metrics between all gray matter regions.

## Resting-state functional MRI (rsMRI)

rsMRI data were acquired using a gradient-echo echo-planar imaging sequence with 8x multislice (multiband) acceleration (TE = 39 ms, TR = 735 ms, 2.4 mm isotropic voxels, 490 volumes, FOV: 88×88×64, flip angle = 52°). Preprocessing followed the UK Biobank pipeline and included correction for susceptibility- and gradient-induced distortions, motion correction using FMRIB's Linear Image Registration Tool (MCFLIRT), grand-mean intensity scaling, high-pass temporal filtering, and structured artefact removal using independent component analysis (ICA) combined with FMRIB's ICA-based X-noiseifier (FIX), with non-neural components discarded. Group ICA was performed using FSL MELODIC to derive spatial maps of resting-state networks, followed by dual regression to estimate subject-specific BOLD time series for each network (*Beckmann et al., 2009*; *Beckmann, 2012*; *Beckmann and Smith, 2004*; *Nickerson et al., 2017*).

To quantify rsMRI, we used 26 IDPs. These included functional connectivity measures – normalized full and partial temporal correlations between node time series – and network amplitudes (standard deviation of network time series) for 21 and 55 independent components (ICs), representing statistically independent or spatially distinct resting-state functional networks, obtained from ICA decompositions at 25 and 100 dimensions, respectively (*Beckmann et al., 2005*; *Cole et al., 2010*; *Lee et al., 2023*). Partial correlations were regularized using ridge regression (*L2* regularization, $\rho$=0.5) to improve estimation stability. All correlation coefficients were Gaussianized (*z*-transformed) and corrected for temporal autocorrelation.

In addition to ICA-based metrics, we derived functional connectomes from parcellated BOLD time series using six combinations of cortical and subcortical atlases, consistent with the structural connectome analysis (*Mansour et al., 2023*). Cortical and subcortical time series, provided separately, were combined prior to connectivity estimation. Parcellation-based functional connectomes were constructed by averaging preprocessed BOLD time series within regions of interest. Atlas labels were resampled from the individual T1-weighted space to the fMRI voxel grid using FreeSurfer's *mri_vol2vol*, and voxel-wise BOLD signals were averaged within each region. Functional connectivity matrices were then computed using the *ConnectivityMeasure* function from the *Nilearn* Python library.

For both ICA- and atlas-based approaches, we complemented correlation-based metrics with tangent space parameterization, defined as deviations of individual connectivity matrices from the group-average connectivity. This involved estimating a group-average covariance matrix from the training data using the Ledoit-Wolf shrinkage estimator and projecting individual covariance matrices from both training and test sets into a common tangent space (*Abbas et al., 2023*; *Dadi et al., 2019*; *Ledoit and Wolf, 2004*; *Ng et al., 2014*; *Pervaiz et al., 2020*). Prior to projection, BOLD time series were *z*-score standardized. The resulting matrices were vectorized, and diagonal elements were excluded.

This approach accounts for the non-Euclidean geometry of covariance and correlation matrices, which lie on a high-dimensional nonlinear curved surface – Riemannian manifold of symmetric positive definite matrices (*Abbas et al., 2023*; *Venkatesh et al., 2020*). Projection into a common tangent space reduces statistical dependencies between covariance estimates and enables the use of conventional linear methods, addressing limitations related to the nonlinear geometry of functional connectivity data and often improving performance relative to standard correlation-based measures (*Abbas et al., 2023*; *Dadi et al., 2019*; *Ng et al., 2014*; *Simeon et al., 2022*; *Venkatesh et al., 2020*).

For each atlas combination, full and partial correlations were computed across the entire sample, while tangent space representations were fitted on the training set and applied to both training and test sets within each cross-validation fold. Pearson correlation coefficients were additionally transformed using the inverse hyperbolic tangent (*arctanh*) function to reduce skewness. Machine learning models were trained separately for each connectivity measure, and the measure with the best training performance was selected for downstream analyses.

## Structural MRI (sMRI)

T1w images were acquired using a 3D MPRAGE sequence (1 mm isotropic voxels, FOV: 208×256×256). Structural preprocessing included gradient distortion correction, brain extraction and defacing, and registration from native space to MNI152 standard space using linear and nonlinear transformations (*Jenkinson et al., 2002*; *Jenkinson et al., 2012*; *Jenkinson and Smith, 2001*; *Mansour et al., 2023*; *Smith, 2002*). Tissue-specific and total brain volumes were estimated using SIENAX-style analysis

(Structural Image Evaluation, using Normalization, of Atrophy: Cross-sectional) (*Smith et al., 2002*). Shapes and volumes of 15 subcortical structures were modeled using FIRST (FMRIB's Integrated Registration and Segmentation Tool) (*Patenaude et al., 2011*). Cortical surface area and thickness were estimated with FreeSurfer, and subcortical segmentations were obtained using the ASEG tool (*Fischl, 2012*).

T2w images were acquired using a 3D SPACE sequence with fluid-attenuated inversion recovery (FLAIR) contrast to enhance visualization of white matter hyperintensities (1.05×1×1 mm voxels, FOV: 192×256×256). T2w images were linearly registered to T1w images using FSL FLIRT (FMRIB's Linear Image Registration Tool), subsequently aligned to MNI space, and bias-field corrected. Total white matter hyperintensity volume (i.e. white matter lesions) was quantified using the BIANCA (Brain Intensity AbNormality Classification Algorithm) (*Griffanti et al., 2016*). Tissue segmentation and bias-field correction were performed in FSL using FAST (FMRIB's Automated Segmentation Tool) (*Zhang et al., 2001*).

To quantify sMRI, we used 20 IDPs, which included gray and white matter volumes and mean intensities, cortical surface area and thickness, volumes of subcortical structures and cerebrospinal fluid, gray-white matter contrast (fractional intensity contrast), and total brain volume. These measures were derived using eight segmentation strategies: FSL FAST regional gray matter volumes (*Zhang et al., 2001*), FSL FIRST subcortical volumes (*Patenaude et al., 2011*), FreeSurfer ASEG subcortical segmentation, FreeSurfer ex-vivo Brodmann area maps, FreeSurfer Destrieux (a2009s) atlas, FreeSurfer Desikan-Killiany-Tourville parcellation, FreeSurfer Desikan-Killiany parcellation (including gray/white matter intensity and pial and white matter surfaces), and FreeSurfer subcortical subsegmentation (*Iglesias et al., 2015*). T2w-derived IDPs included total volumes of deep white matter, periventricular white matter, and white matter hyperintensities (*Supplementary file 4*).

## MRI confounds

We adjusted all MRI modalities for common and modality-specific confounds as described by *Alfaro-Almagro et al., 2021*. Common confounds included scanning site, acquisition date, head size, scanner table position, radiofrequency receive coil position, brain position within the scanner, mismatch between T1w images and the standard-space template, and structural motion (head motion estimated from T1w images) (*Alfaro-Almagro et al., 2021*). Head size was defined as a volumetric scaling factor obtained during transformation from native T1w space to MNI space using SIENAX (*Alfaro-Almagro et al., 2018*; *Smith et al., 2002*). Scanning site (four UK Biobank centres) was encoded using dummy variables, and the acquisition date was converted to Unix time. Modality-specific confounds included head motion (mean displacement between consecutive time points, averaged across the brain and time), intensity scaling, and mismatch between non-structural images and the T1w reference. Motion summaries additionally included median absolute and relative displacement (*Supplementary file 4*).

For dwMRI, motion estimates were obtained using FSL *eddy*. To account for slow signal drifts (e.g. heating effects), we included additional confounds related to advanced eddy current correction, such as expanded search space parameters. We also included the number of slices identified as outliers by *eddy* as a motion-related confound.

For rsMRI, motion parameters were estimated using FSL FEAT (FMRI Expert Analysis Tool). Additional motion-related confounds were derived using DVARS, which decomposes signal variability into fast (D-var), slow (S-var), and edge (E-var) components (*Afyouni and Nichols, 2018*). D-var represents the mean squared difference between consecutive volumes, S-var reflects the mean squared average of adjacent volumes, and E-var captures edge-related variability at the beginning and end of the time series. DVARS-based confounds included the mean, median, and 90th percentile of S-var and D-var, normalized by total signal variance (A-var).

All confounds were regressed out from MRI features following standardization of both confounds and features. Interactive visualizations of variance explained by confound groups are available at https://www.fmrib.ox.ac.uk/ukbiobank/confounds/plots_2020_03_11/index.html.

## Data analysis

### Machine learning

To model the relationship between mental health and cognition, we employed PLSR to predict the *g*-factor from 133 mental health variables, and to model the relationship between brain and cognition,

we used a two-step stacking approach (*Krämer et al., 2024*; *Pat et al., 2022*; *Tetereva et al., 2022*; *Tetereva et al., 2025*) that integrates information from neuroimaging phenotypes across three MRI modalities. In the first step, we trained 72 base (first-level) PLSR models, each predicting the *g*-factor from a single neuroimaging phenotype. In the second step, we used the predicted values from these base models as input features for stacked models, which again predicted the *g*-factor. We constructed four stacked models based on the source of the base predictions: one each for dwMRI ('dwMRI Stacked'), rsMRI ('rsMRI Stacked'), sMRI ('sMRI Stacked'), and a combined model incorporating all modalities ('All MRI Stacked'). Each stacked model was trained using one of four machine learning algorithms – ElasticNet, Random Forest, XGBoost, or Support Vector Regression (SVR) – selected individually for each model. Analyses were performed in *Python* (version 3.11), and machine learning models were implemented using the *scikit-learn* library.

We employed nested cross-validation to predict cognition from mental health indices and neuro-imaging phenotypes (*Figure 1*). Nested cross-validation is a robust method for evaluating machine learning models while tuning their hyperparameters, ensuring that performance estimates are both accurate and unbiased. Here, we used a nested cross-validation scheme with five outer folds and ten inner folds (*Wilimitis and Walsh, 2023*). We started by dividing the entire dataset into five outer folds. Each fold took a turn being held out as the outer-fold test set (20% of the data), while the remaining fourfolds (80% of the data) were used as an outer-fold training set. Within each outer-fold training set, we performed a second layer of cross-validation – this time splitting the data into ten inner folds. These inner folds were used exclusively for hyperparameter tuning: models were trained on nine of the inner folds and validated on the remaining one, cycling through all ten combinations.

We then selected the hyperparameter configuration that minimized the mean squared error (*MSE*) across inner-fold validation sets. The model was then retrained on the full outer-fold training set using the optimal configuration and evaluated on the outer-fold test set using Pearson correlation (*r*), the coefficient of determination ($R^2$), mean absolute error (*MAE*), and *MSE* for the predicted and observed *g*-factor values. The evaluation metrics were computed as follows:

$$R^2 = 1 - \frac{\sum_{i=1}^{n}(y_i - \hat{y}_i)^2}{\sum_{i=1}^{n}(y_i - y_i')^2},$$

$$r = \frac{\sum_{i=1}^{n}(y_i - y_i')(\hat{y}_i - \hat{y}_i')}{\sqrt{\sum_{i=1}^{n}(y_i - y_i')^2 \sum_{i=1}^{n}(\hat{y}_i - \hat{y}_i')^2}},$$

$$MAE = \frac{1}{n}\sum_{i=1}^{n}\left|y_i - \hat{y}_i\right|,$$

$$MSE = \frac{1}{n}\sum_{i=1}^{n}(y_i - \hat{y}_i)^2,$$

where $y_i$, $y_{ii}'$, and $\hat{y}i$ denote observed values, the mean of observed values, and predicted values, respectively.

Performance metrics were averaged across the five outer folds for each feature set. For rsMRI IDPs, the choice of functional connectivity quantification method – full correlation, partial correlation, or tangent space parameterization – was treated as a hyperparameter. The method yielding the best performance in the inner cross-validation (training) folds was selected and subsequently evaluated on the outer-fold test sets for predicting the *g*-factor.

This entire process was repeated for each of the five outer folds, ensuring that every data point is used for both training and testing, but never at the same time. We opted for five outer folds instead of ten to reduce computational demands, particularly memory and processing time, given the substantial volume of neuroimaging data involved in model training. Five outer folds led to an outer-fold test set at least n=4000, which should be sufficient for model evaluation. In contrast, we retained ten inner folds to ensure robust and stable hyperparameter tuning, maximising the reliability of model selection.

The first-level machine learning model applied to mental health and MRI data was PLSR. PLSR is a multivariate method that simultaneously performs dimensionality reduction and regression by projecting correlated predictor variables onto a set of latent components that maximise covariance

with the response variable(s) (**Wold et al., 2001**). This makes it particularly suitable for high-dimensional and collinear neuroimaging and behavioural data.

Formally, PLSR decomposes the predictor matrix $X$ into orthogonal latent scores and loadings and models the response variable $Y$ as a linear function of these components, yielding a regression model of the form:

$$Y = XB + F,$$

where $B$ represents regression coefficients and $F$ residuals.

This is achieved by projecting the original predictors onto a set of latent variables ($X$-scores), which are weighted linear combinations of the original variables:

$$X = TP' + E, \text{ with } T = XW^*,$$

where $T$ is the matrix of latent scores, $P$ the loadings, $W^*$ the transformed (rotated) weight matrix, and $E$ the residual matrix. The scores $T$ represent projections of the observations onto the latent components, whereas the loadings $P$ describe how the original variables contribute to these components and are represented in the latent space. The weights determine how the original variables are combined to form the latent scores. Specifically, weights are estimated by maximising the covariance between $X$ and $Y$, thereby capturing the most predictive variation in the data, and are subsequently normalized so that the latent scores have unit variance.

The response variable is then modeled as a function of the latent scores:

$$Y = TC' + F,$$

where $C$ is the matrix of $Y$-weights (with columns $c_a$) that relate the latent components to the response variable(s). In the case of multiple response variables, PLSR similarly defines $Y$-scores $U=YC$, such that $Y=UC'+G$, where $U$ represents projections of $Y$ onto the latent space. This leads to the regression formulation:

$$Y = XB + F,$$

where $B=W^*C'$. In this framework, the latent components act as orthogonal predictors, reducing multi-collinearity while preserving information relevant for predicting $Y$.

In practice, PLSR iteratively extracts components, with each step followed by deflation of the predictor matrix to ensure that subsequent components capture new, orthogonal sources of covariance between $X$ and $Y$ (**Wold et al., 2001**). This deflation procedure ensures that the extracted scores are orthogonal, while weights remain orthonormal across components. Thus, the latent variables provide a low-dimensional representation of the predictors optimized for the prediction of the response.

Geometrically, $X$-scores ($t_a$, i.e. the score vectors corresponding to the columns of the score matrix $T$) can be interpreted as coordinates of the projection of the predictor matrix $X$ onto an $A$-dimensional latent hyperplane that approximates the original data. Each component defines a direction within this space, with $X$-loadings ($p_a$, i.e. the loading vectors corresponding to the columns of the loading matrix $P$) representing the corresponding direction coefficients (slopes) that describe how the original variables contribute to these components. These directions are chosen to maximise the covariance between $X$ and $Y$, thereby defining the latent space that best captures their shared structure. A comprehensive overview of the mathematical and geometric interpretation of PLSR is available at: https://learnche.org/pid/latent-variable-modelling/projection-to-latent-structures/index.

In summary, PLSR constructs new predictor variables ($X$-scores) that (i) represent the original predictors in a latent space, (ii) are linear combinations of the original variables, and (iii) are maximally predictive of the response variable. The only hyperparameter in PLSR is the number of components. We optimized this parameter using nested cross-validation with random shuffling. The optimal number of components was selected via grid search (*GridSearchCV*, *scikit-learn*) by minimising the *MAE*. The final model was retrained on the full outer-fold training set using the optimal number of components and evaluated on the held-out test set to obtain out-of-sample predictions of the *g*-factor.

To train the second-level models, we evaluated four machine learning algorithms – ElasticNet, Random Forest, XGBoost, and SVR – and selected the model with the highest out-of-sample $R^2$ (*Chicco et al., 2021*). The hyperparameter grids for all models are provided in *Supplementary file 5*.

## ElasticNet

ElasticNet is a linear regression model that combines *L*1 (Lasso) and *L*2 (Ridge) regularization to address multicollinearity and improve generalization (*Hoerl, 2020*; *Kotz et al., 1982*; *Tibshirani, 1996*; *Zou and Hastie, 2005*). The *L*1 penalty promotes sparsity by shrinking some coefficients to zero (enabling variable selection), whereas the *L*2 penalty stabilises estimates by shrinking coefficients toward zero without eliminating them. The mixing parameter (*l1_ratio*) controls the relative contribution of *L*1 and *L*2 penalties (0 = Ridge, 1 = Lasso), and the regularization strength (*alpha*) controls the overall degree of shrinkage. This combination allows ElasticNet to retain groups of correlated predictors while reducing overfitting.

Thus, ElasticNet addresses several limitations of Lasso regression: it can handle settings with more predictors than observations; it avoids selecting only a single variable from a group of correlated predictors; and it improves prediction accuracy in the presence of multicollinearity (*van Erp et al., 2019*). Furthermore, by combining *L*1 and *L*2 regularization, ElasticNet promotes more parsimonious models while mitigating the redundancy characteristic of Ridge regression, which tends to retain all correlated predictors through continuous shrinkage (*Zou and Hastie, 2005*).

## Support vector regression (SVR)

SVR extends Support Vector Machines (SVM), a supervised learning method introduced by Vapnik, to regression problems (*Awad and Khanna, 2015*; *López et al., 2022*; *Rodríguez-Pérez and Bajorath, 2022*; *Vapnik, 1998*; *Vapnik, 1995*). SVMs can capture nonlinear relationships and handle high-dimensional data, including settings with many predictors and relatively few observations, by implicitly mapping the original data into a higher-dimensional feature space using the kernel trick. This approach computes inner products between observations in the transformed space without explicitly performing the transformation, enabling efficient modeling of complex relationships.

In the SVM framework, the goal is to identify an optimal hyperplane that maximises the margin between data points. The margin is defined by the distance between the hyperplane and the closest data points, known as support vectors (SVs), which determine the position and orientation of the decision boundary. In SVR, this concept is adapted to regression through the introduction of an $\varepsilon$-insensitive tube around the regression function. Errors within this tube are not penalized, whereas deviations exceeding $\varepsilon$ (the distance from boundaries to the hyperplane) are penalized via an $\varepsilon$-insensitive loss function. The model, therefore, seeks a function that is both as flat as possible and captures the majority of the data within this tolerance region. Support vectors in SVR are the observations that lie outside the $\varepsilon$-tube and thus define the shape and position of the regression function.

To capture nonlinear relationships, we used the radial basis function (RBF) kernel, which measures similarity between observations based on their distance in the input space. In other words, in the regression setting, the RBF kernel enables the estimation of similarity between new observations and support vectors, which define the shape and position of the regression function (*Elish, 2014*; *Ramedani et al., 2014*; *Smola and Schölkopf, 2004*). It maps the data into a higher-dimensional feature space where a linear relationship can approximate complex nonlinear patterns in the original space (*Chen and Bakshi, 2009*; *Ramedani et al., 2014*). This allows SVR to model nonlinear dependencies by effectively 'linearising' them after transformation.

The kernel width parameter ($\gamma$) controls how rapidly this similarity decays with distance, thereby determining the strength of the influence of a single training instance. Larger $\gamma$ values make the model focus more strongly on nearby observations (i.e. highly local influence), resulting in more flexible but potentially overfitted solutions, whereas smaller $\gamma$ values lead to broader, smoother patterns that generalise better.

The regularization parameter (*C*) controls the trade-off between fitting the training data and limiting model complexity. Higher values of *C* prioritisze minimizing prediction errors, allowing the model to fit the training data more closely, whereas lower values tolerate more errors but yield simpler models that generalize better to unseen data (*Awad and Khanna, 2015*; *Balfer and Bajorath, 2015*; *Liu and*

*Zheng, 2006*; *Rodríguez-Pérez and Bajorath, 2022*). In practical terms, $C$ determines how strongly the model penalises deviations outside the $\varepsilon$-insensitive tube.

## Random forest

Random Forest is an ensemble learning method within the Classification and Regression Tree (CART) framework that constructs multiple decision trees on bootstrap samples of the data and aggregates their predictions (*Breiman, 2001*; *Breiman et al., 2017*; *Schonlau and Zou, 2020*; *Svetnik et al., 2003*). It employs two sources of randomness – bootstrap sampling of observations and random selection of predictor subsets at each split (*max_features*, also referred to as *mtry*) – to decorrelate trees, reduce variance, and improve generalization relative to a single decision tree. Each tree is typically grown to substantial depth (constrained by *max_depth* and minimum node size), allowing the model to capture complex nonlinear relationships and interactions. Predictions are obtained by averaging outputs across trees, which stabilises estimates and smooths nonlinearities.

Splits within each tree are selected to maximize the reduction in impurity, ensuring greater homogeneity of the resulting nodes. In regression settings, impurity is measured by the residual sum of squares, and splits are chosen to minimize within-node variance. The magnitude of impurity reduction also provides a basis for estimating feature importance, with larger reductions indicating greater predictive contribution (*Breiman et al., 2017*; *Ishwaran, 2015*; *Nembrini et al., 2018*).

Key hyperparameters include the number of decision trees built on different bootstrap samples (*n_estimators*), which controls the size of the ensemble; the maximum tree depth (*max_depth*), which determines the number of splits within each tree, governing model complexity; and the number of predictors considered at each split (*max_features*).

## XGBoost

XGBoost (eXtreme Gradient Boosting) is a gradient boosting framework that sequentially builds decision trees to minimise a specified loss function, i.e., the discrepancy between observed and predicted values of the response variable, using gradient descent (*Chen and Guestrin, 2016*; *Tarwidi et al., 2023*). Each new tree is fitted to the residuals of the previous ensemble, progressively improving model performance. In other words, the model iteratively corrects errors made by earlier trees. XGBoost incorporates regularization of tree complexity and feature weights to reduce overfitting and enhance generalizability.

## Standardization and statistical significance

To prevent data leakage, we standardized the data using the mean and standard deviation derived from the training set and applied these parameters to the corresponding test set within each outer fold. This standardization was performed at three key stages: before *g*-factor derivation, before regressing out modality-specific confounds from the MRI data, and before stacking. Similarly, to maintain strict separation between training and testing data, both base and stacked models were trained exclusively on participants from the outer-fold training set and subsequently applied to the corresponding outer-fold test set.

To assess the statistical significance of the models, we aggregated the predicted and observed *g*-factor values from each outer-fold test set. We then computed a bootstrap distribution of Pearson's correlation coefficient (*r*) by resampling with replacement 5000 times, generating 95% confidence intervals (CIs). Model performance was considered statistically significant if the 95% CI did not include zero, indicating that the observed associations were unlikely to have occurred by chance.

## Feature importance

To identify the neuroimaging features that contribute most to the predictive performance of top-performing phenotypes within each modality, while accounting for the potential latent components derived from neuroimaging, we assessed feature importance using the Haufe transformation (*Haufe et al., 2014*). Specifically, we calculated Pearson correlations between the predicted *g*-factor and scaled and centred neuroimaging features across five outer-fold test sets. We also examined whether the performance of neuroimaging phenotypes in predicting cognition per se is related to their ability to explain the link between cognition and mental health. Here, we computed the correlation between

the predictive performance of each neuroimaging phenotype and the proportion of the cognition–mental health relationship it captures.

## Commonality analysis

Finally, we conducted a series of commonality analyses to quantify the proportion of the cognition–mental health relationship captured by MRI-based neural indicators (*Nimon et al., 2008*; *Seibold and McPHEE, 1979*; *Viswesvaran, 1998*). For each set of MRI data (i.e. individual neuroimaging phenotypes, neuroimaging phenotypes stacked within each MRI modality, and neuroimaging phenotypes stacked across all MRI modalities), for both observed (i.e. derived directly from cognitive scores, not mental health or MRI data) and predicted *g*-factors (i.e. predicted from either mental health indices or neuroimaging phenotypes using the top-performing algorithm), we separately pooled values across all five outer-fold test sets to reconstruct the complete dataset. We then applied a series of linear regression models, treating the observed *g*-factor as a response variable and the *g*-factor predicted from mental health (mental health-*g*) and neuroimaging phenotypes (neuroimaging-*g*) as two separate explanatory variables:

$$\textbf{Model 1: } g_{\text{observed}} \sim \hat{g}_{\text{mental health}}$$

$$\textbf{Model 2: } g_{\text{observed}} \sim \hat{g}_{\text{neuroimaging}}$$

$$\textbf{Model 3: } g_{\text{observed}} \sim \hat{g}_{\text{mental health}} + \hat{g}_{\text{neuroimaging}}$$

Next, we decomposed the total variance of the observed *g*-factor, $R^2$, into the variance explained uniquely or commonly by mental health-*g* and neuroimaging-*g*, as follows *Nimon et al., 2008*; *Seibold and McPHEE, 1979*; *Wang et al., 2025*:

$$\textbf{Unique}_{\textbf{mental health}} = R^2_{\text{mental health, neuroimaging}} - R^2_{\text{neuroimaging}}$$

$$\textbf{Unique}_{\textbf{neuroimaging}} = R^2_{\text{mental health, neuroimaging}} - R^2_{\text{mental health}}$$

$$\textbf{Common}_{\textbf{mental health, neuroimaging}} = R^2_{\text{mental health, neuroimaging}} - \text{Unique}_{\text{mental health}}$$
$$- \text{Unique}_{\text{neuroimaging}} = R^2_{\text{mental health}} + R^2_{\text{neuroimaging}} - R^2_{\text{mental health, neuroimaging}}$$

We quantified the contribution of each neuroimaging phenotype to explaining the cognition–mental health relationship by a percentage ratio between (a) the common effects between mental health-*g* and neuroimaging-*g* and (b) the total effects of mental health-*g*. That is, the numerator is the common variance that neuroimaging-*g* shares with mental health-*g* in explaining the observed *g*-factor, and the denominator is the variance of mental health-*g* that explains the observed *g*-factor regardless of neuroimaging-*g*:

$$\% \textbf{ Cognition–mental health relationship explained by the brain} = \frac{Common_{mental\ health,\ neuroimaging}}{R^2_{mental\ health}},$$

where $R^2_{\text{mental health}} = \text{Common}_{\text{mental health, neuroimaging}} + \text{Unique}_{\text{mental health}}$.

Each commonality analysis was conducted independently and, therefore, included a different number of participants, depending on how many were available for a given neuroimaging phenotype or modality.

To understand how demographic factors, including age and sex, contribute to this relationship, we also conducted a separate set of commonality analyses treating age, sex, $age^2$, age×sex, and $age^2$×sex as an additional set of explanatory variables.

## Acknowledgements

We are thankful to Professor Bruce Russell for his assistance with access to the UK Biobank dataset. NP and JD were supported by Health Research Council of New Zealand (grant numbers 21/618 and 24/838) and by Neurological Foundation of New Zealand (grant number 2350 PRG). NP was supported by the Ministry of Business, Innovation and Employment (grant numbers UOA2421 and RTVU2403). IB was supported by the University of Otago.

## Additional information

### Funding

| Funder | Grant reference number | Author |
|---|---|---|
| Health Research Council of New Zealand | 21/618 | Jeremiah D Deng<br>Narun Pat |
| Neurological Foundation of New Zealand | 2350 PRG | Jeremiah D Deng<br>Narun Pat |
| Ministry of Business, Innovation and Employment | UOA2421 | Narun Pat |
| University of Otago | The University of Otago Doctoral Scholarship | Irina Buianova |
| Health Research Council of New Zealand | 24/838 | Jeremiah D Deng<br>Narun Pat |
| Ministry of Business, Innovation and Employment | RTVU2403 | Narun Pat |

The funders had no role in study design, data collection and interpretation, or the decision to submit the work for publication.

### Author contributions

Irina Buianova, Conceptualization, Data curation, Formal analysis, Validation, Investigation, Visualization, Methodology, Writing – original draft, Writing – review and editing; Mateus Silvestrin, Software, Methodology; Jeremiah D Deng, Methodology; Narun Pat, Conceptualization, Resources, Supervision, Funding acquisition, Methodology, Project administration, Writing – review and editing

### Author ORCIDs

Irina Buianova ⓘ https://orcid.org/0000-0001-9105-3172
Mateus Silvestrin ⓘ https://orcid.org/0000-0002-3482-3676
Jeremiah D Deng ⓘ https://orcid.org/0000-0003-3727-4403
Narun Pat ⓘ https://orcid.org/0000-0003-1459-5255

### Ethics

This research involved secondary analysis of data from the UK Biobank resource. All UK Biobank participants provided informed consent directly to UK Biobank at recruitment, and separate consent for this study was not required. The study was conducted under UK Biobank Application Number 70132. UK Biobank has ethical approval from the North West Multi-centre Research Ethics Committee (reference 16/NW/0274) as a Research Tissue Bank, and all analyses were performed in accordance with the relevant guidelines and regulations.

Reviewer #1 (Public review): https://doi.org/10.7554/eLife.108109.3.sa1
Reviewer #2 (Public review): https://doi.org/10.7554/eLife.108109.3.sa2
Author response https://doi.org/10.7554/eLife.108109.3.sa3

## Additional files

### Supplementary files
MDAR checklist

Supplementary file 1. The unique variance of each cognitive score that is not explained by the *g*-factor, the variance of each score explained by the *g*-factor, and the proportion of covariance in cognitive scores captured by the *g*-factor.

Supplementary file 2. Out-of-sample predictive performance of dwMRI in the PLSR model averaged across five folds.

Supplementary file 3. Whole-sample distributions of mental health measures used as features in machine learning models (N = 21,077).

Supplementary file 4. UK Biobank neuroimaging variables included in the study.

Supplementary file 5. Hyperparameter grids for machine learning algorithms used in the study.

## Data availability

This study used data from the UK Biobank resource (Application No. 70132). These data cannot be publicly shared by the authors due to legal and ethical restrictions imposed by UK Biobank. Access to individual-level data is governed by UK Biobank's data access policies, which are designed to protect participant confidentiality. Researchers can access the data by submitting an application directly to UK Biobank (https://www.ukbiobank.ac.uk/enable-your-research/apply-for-access). Applications must include a research proposal outlining the scientific rationale and intended use of the data, and are reviewed as part of UK Biobank's access procedures. Access may be granted to approved researchers from academic, charity, government, and commercial organisations, subject to UK Biobank's terms and conditions. Due to these restrictions, we are not permitted to share raw or deidentified individual-level data. Redistribution of such data is contractually prohibited, and deidentification does not eliminate the risk of participant re-identification. This manuscript is a computational study and did not generate new primary data. Numerical data underlying the figures are provided as figure source data files. All analyses can be reproduced using the code provided by the authors. The modelling and analysis code is openly available on GitHub: https://github.com/HAM-lab-Otago-University/UKBiobank/ (copy archived at *Buianova, 2026*).

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

# Appendix 1

**Appendix 1—table 1.** Derivation of mental health scores.

| Disorder/Exposure | Definition | Fields | Resources |
|---|---|---|---|
| PHQ-9 | The sum of the nine depressive symptoms scored 0–4:<br>Little interest or pleasure in doing things<br>Feeling down depressed, or hopeless<br>Trouble sleeping<br>Feeling tired<br>Poor appetite or overeating<br>Feeling bad about yourself<br>Trouble concentrating<br>Moving or speaking slowly or fidgety or restless<br>Thoughts that you would be better off dead<br>Answers "Prefer not to answer" were assigned the lowest score (0). | 20507<br>20508<br>20510<br>20511<br>20513<br>20514<br>20517<br>20518<br>20519 | *Davis et al., 2020*<br>*Kroenke et al., 2010* |
| Depression ever | At least one core symptom of depression (Persistent sadness or Loss of interest) that lasted most or all of the day on most or all days within two weeks with some or a lot of impact on normal activity, plus additional depressive symptoms that represent a change in the mental and/or physical state from usual state and occur over the same period with thoughts about death. A total of ≥5 symptoms, including core symptoms. The score is obtained based on the DSM definition of major depressive disorder. | 20435<br>20436<br>20437<br>20439<br>20440<br>20441<br>20446<br>20449<br>20450<br>20532<br>20536 | *Davis et al., 2020*<br>CIDI-SF (Composite International Diagnostic Interview – Short Form), depression module, lifetime version<br>*Kessler et al., 1998* |
| Bipolar affective disorder type I | Ever had depression and ever manic/hyper or irritable, plus at least three manifestations of mania or irritability (more talkative, more restless, thoughts racing, needed less sleep, more creative or had more ideas, easily distracted, more confident, more active) or four manifestations if never manic/hyper, plus duration of symptoms for a week or more and symptoms caused significant problems. | 20435<br>20436<br>20437<br>20439<br>20440<br>20441<br>20446<br>20449<br>20450<br>20492<br>20493<br>20501<br>20502<br>20532<br>20536<br>20548 | *Davis et al., 2020*<br>*Cerimele et al., 2014*<br>*Carvalho et al., 2015* |
| Bipolar affective disorder type II | Ever had depression and ever manic/hyper or irritable, plus at least three manifestations of mania or irritability (more talkative, more restless, thoughts racing, needed less sleep, more creative or had more ideas, easily distracted, more confident, more active) or four manifestations if never manic/hyper, plus duration of symptoms for a week or more and symptoms did not cause significant problems. | 20435<br>20436<br>20437<br>20439<br>20440<br>20441<br>20446<br>20449<br>20450<br>20492<br>20501<br>20502<br>20532<br>20536<br>20548 | *Davis et al., 2020* |

*Appendix 1—table 1 Continued on next page*

*Appendix 1—table 1 Continued*

| Disorder/ Exposure | Definition | Fields | Resources |
|---|---|---|---|
| Subthreshold depressive symptoms ever | Does not meet Composite International Diagnostic Interview (CIDI) diagnostic criteria for depression, but has at least one of the following symptoms:<br>Persistent depression or anhedonia based on CIDI<br>PHQ9 score for current depressive symptoms exceeds the threshold for mild depression<br>The presence of a clinician diagnosis of depression | 20002<br>20435<br>20436<br>20437<br>20439<br>20440<br>20441<br>20446<br>20449<br>20450<br>20507<br>20508<br>20510<br>20511<br>20513<br>20514<br>20517<br>20518<br>20519<br>20532<br>20536<br>20544 | *Davis et al., 2020*<br>National Institute for Health and Clinical Excellence. Depression in adults: recognition and management. NICE Clinical Guideline CG90 |
| Depression single episode | A single episode of depression without bipolar disorder type I. | 20435<br>20436<br>20437<br>20439<br>20440<br>20441<br>20442<br>20446<br>20449<br>20450<br>20492<br>20493<br>20501<br>20502<br>20532<br>20536 | *Davis et al., 2020* |
| Recurrent depression | More than one episode of depression throughout a lifetime without bipolar disorder type I. | 20435<br>20436<br>20437<br>20439<br>20440<br>20441<br>20442<br>20446<br>20449<br>20450<br>20492<br>20493<br>20501<br>20502<br>20532<br>20536 | *Davis et al., 2020* |

*Appendix 1—table 1 Continued on next page*

*Appendix 1—table 1 Continued*

| Disorder/ Exposure | Definition | Fields | Resources |
|---|---|---|---|
| Depression single episode triggered by a loss | A single episode of depression that started within two months after a traumatic event. | 20435 20436 20437 20439 20440 20441 20442 20446 20447 20449 20450 20492 20493 20501 20502 20532 20536 | *Davis et al., 2020* |
| Current depression | At least a single episode of depression ('Depression ever') with a minimum of 5 depression symptoms from the PHQ-9 occurring more than half days or for several days for suicidal thoughts. | 20435 20436 20437 20439 20440 20441 20446 20449 20450 20507 20508 20510 20511 20513 20514 20517 20518 20519 20532 20536 | *Davis et al., 2020* *Manea et al., 2012* |
| Current severe depression | At least a single episode of depression ('Depression ever') As current depression (above) with PHQ score >15. | 20435 20436 20437 20439 20440 20441 20446 20449 20450 20507 20508 20510 20511 20513 20514 20517 20518 20519 20532 20536 | *Davis et al., 2020* *Manea et al., 2012* |

*Appendix 1—table 1 Continued on next page*

*Appendix 1—table 1 Continued*

| Disorder/ Exposure | Definition | Fields | Resources |
|---|---|---|---|
| GAD-7 | The sum of the recent symptoms of anxiety scored 0–3:<br>Feelings of nervousness or anxiety<br>Inability to stop or control worrying<br>Worrying too much about different things<br>Trouble relaxing<br>Restlessness<br>Easy annoyance or irritability<br>Feelings of foreboding | 20505<br>20506<br>20509<br>20512<br>20515<br>20516<br>20520 | *Davis et al., 2020*<br>*Kroenke et al., 2010* |
| Lifetime anxiety disorder (GAD ever) | Ever felt worried, tense, or anxious for most of the day for at least six months with difficulties in controlling symptoms. The symptoms were often difficult to control, they interfered with daily activity and were accompanied by at least three somatic symptoms (restless, keyed up or on edge., easily tired, difficulty keeping the mind on current activity, more irritable than usual, tense muscles, trouble falling or staying asleep). | 20417<br>20418<br>20419<br>20420<br>20421<br>20422<br>20423<br>20425<br>20426<br>20427<br>20429<br>20537<br>20538<br>20539<br>20540<br>20541<br>20542<br>20543 | *Davis et al., 2020*<br>CIDI-SF (Composite International Diagnostic Interview – Short Form), GAD module, lifetime version. Scored based on the DSM definition of GAD<br>*Gigantesco and Morosini, 2008*<br>*Kessler et al., 1998*<br>National Institute for Health and Clinical Excellence. Generalised anxiety disorder and panic disorder in adults: management. NICE Clinical Guideline CG113 |
| Current anxiety | Ever had GAD ('GAD Ever') and GAD-7 score ≥10. Subdivided into mild, moderate, and severe with cut-offs at 5, 10, and 15 | 20417<br>20418<br>20419<br>20420<br>20421<br>20422<br>20423<br>20425<br>20426<br>20427<br>20429<br>20505 | *Davis et al., 2020*<br>*Kroenke et al., 2010* |
| Current anxiety | Ever had GAD ('GAD Ever') and GAD-7 score ≥10. Subdivided into mild, moderate, and severe with cut-offs at 5, 10, and 15. | 20506<br>20509<br>20512<br>20515<br>20516<br>20520<br>20537<br>20538<br>20539<br>20540<br>20541<br>20542<br>20543 | *Davis et al., 2020*<br>*Kroenke et al., 2010* |

*Appendix 1—table 1 Continued on next page*

*Appendix 1—table 1 Continued*

| Disorder/ Exposure | Definition | Fields | Resources |
|---|---|---|---|
| N-12 | The sum of the following scores:<br>Mood swings<br>Miserableness<br>Irritability<br>Sensitivity/hurt feelings<br>Fed-up feelings<br>Nervous feeling<br>Worrier/anxious feelings<br>Tense/'highly strung'<br>Worry too long after embarrassment<br>Suffer from 'nerves'<br>Loneliness, isolation<br>Guilty feelings | 1920<br>1930<br>1940<br>1950<br>1960<br>1970<br>1980<br>1990<br>2000<br>2010<br>2020<br>2030 | *Dutt et al., 2022*<br>*Smith et al., 2013a* |
| PDS | Ever been depressed or unenthusiastic for at least one week and seen either a GP or psychiatrist for nerves, anxiety, tension, or depression. | 2090<br>2100<br>4598<br>4609<br>4631<br>5375 | *Dutt et al., 2022* |
| RDS-4 | Frequency of depressed mood, disinterest, restlessness, and tiredness during the past two weeks scored 1–4. | 2050<br>2060<br>2070<br>2080 | *Dutt et al., 2022* |
| PCL-6 | The sum of scores on the core symptoms of PTSD in the past month:<br>Repeated disturbing thoughts of a stressful experience<br>Felt very upset when reminded of a stressful experience<br>Avoided activities or situations because of a previous stressful experience<br>Felt distant from other people<br>Felt irritable or had angry outbursts in the past month<br>Recent trouble concentrating on things<br>The symptoms are grouped into three clusters:<br>Memories, thoughts, or images, upset when reminded<br>Avoid activities or situations, feeling distance or cut-off<br>Irritable or angry, difficulty concentrating | 20494<br>20495<br>20496<br>20497<br>20498<br>20508 | *Davis et al., 2020*<br>*Lang and Stein, 2005* |
| PTSD | PCL-6 score ≥14. | 20494<br>20495<br>20496<br>20497<br>20498<br>20508 | *Davis et al., 2020* |
| Unusual experience | Experience of hallucinations or delusions, such as:<br>Unreal voice<br>Unreal vision<br>Believed in an unreal conspiracy against self<br>Believed in unreal communications or signs | 20463<br>20468<br>20471<br>20474 | *Davis et al., 2020*<br>*Nuevo et al., 2012* |
| Recent unusual experience | Reports at least one or two hallucination or delusion episodes within the last year. | 20467 | *Davis et al., 2020* |
| Life not worth living | Ever felt that life was not worth living. | 20479 | *Davis et al., 2020* |
| Self-harm | Ever harmed self, whether or not meant to die. | 20480 | *Davis et al., 2020* |
| Non-suicidal self-harm | Ever self-harmed without intention to end life, i.e., never attempted suicide. | 20480<br>20483 | *Davis et al., 2020* |
| Suicide attempt | Ever harmed self with intent to end life. | 20480<br>20483 | *Davis et al., 2020* |

*Appendix 1—table 1 Continued on next page*

*Appendix 1—table 1 Continued*

| Disorder/ Exposure | Definition | Fields | Resources |
|---|---|---|---|
| AUDIT | The sum of scores (0–4) on questions about alcohol consumption comprising three domains: Consumption: Frequency, amount of typical drinks, frequency of having six or more drinks Dependence: Unable to stop, failed to do what expected due to drinking, needed to drink first thing Harm: Guilt due to drinking, unable to remember due to drink Plus had injuries due to drinking or advice to cut down on drinking. | 20403 20405 20407 20408 20409 20411 20412 20413 20414 20416 | *Davis et al., 2020 Saunders et al., 1993 Reinert and Allen, 2007* |
| Alcohol consumption (AUDIT-C) | Sum of questions 1–3 of the Alcohol Consumption domain. | 20403 20414 20416 | *Sanchez-Roige et al., 2019* |
| Problems caused by alcohol (AUDIT-P) | Sum of questions 4–10 of the Alcohol Dependence and Alcohol Harm domains. | 20405 20407 20408 20409 20411 20412 20413 | *Sanchez-Roige et al., 2019* |
| Hazardous/ Harmful alcohol use | AUDIT score ≥8. | 20403 20405 20407 20408 20409 20411 20412 20413 20414 20416 | *Davis et al., 2020 Babor et al., 2001 Stansfeld et al., 2016* |
| Current alcohol dependence | AUDIT score ≥15. | 20403 20405 20407 20408 20409 20411 20412 20413 20414 20416 | *Davis et al., 2020 Babor et al., 2001 Drummond et al., 2016* |
| Alcohol dependence ever | Ever physically dependent on alcohol. | 20404 | *Davis et al., 2020* |
| Addiction ever | Ever addicted to any substance or behaviour. | 20401 | *Davis et al., 2020* |
| Substance addiction | Ever been addicted to alcohol, illicit/recreational drugs, or medication. | 20406 20456 20503 | *Davis et al., 2020* |
| Current addiction | Ongoing addiction or dependence. | 20415 20432 20457 20504 | *Davis et al., 2020* |
| Cannabis ever | Taking cannabis at least once in life. | 20453 | *Davis et al., 2020* |
| Cannabis daily | Maximum frequency of taking cannabis when using it every day. | 20454 | *Davis et al., 2020* |

*Appendix 1—table 1 Continued on next page*

*Appendix 1—table 1 Continued*

| Disorder/ Exposure | Definition | Fields | Resources |
|---|---|---|---|
| Childhood adverse events | A positive score if any of the five questions of the Childhood Trauma Screen (CTS) reach the threshold:<br>Felt loved as a child ≤3 (never, rarely, or sometimes)<br>Physically abused by family as a child ≥2 (often or very often)<br>Felt hated by a family member as a child ≥2 (often or very often)<br>Sexually molested as a child ≥2 (often or very often)<br>Someone to take to the doctor when needed as a child ≤4 (never, rarely, sometimes, or often) | 20487<br>20488<br>20489<br>20490<br>20491 | *Davis et al., 2020*<br>*Walker et al., 1999* |
| Adult adverse events | A positive score if any of the five questions of the Adult Trauma Screen reach the threshold:<br>Been in a confiding relationship as an adult ≤3 (never, rarely, or sometimes)<br>Physical violence by partner or ex-partner as an adult ≥2 (often or very often)<br>Belittlement by partner or ex-partner as an adult ≥2 (often or very often)<br>Sexual interference by partner or ex-partner without consent as an adult ≥2 (often or very often)<br>Able to pay rent/mortgage ≤4 (never, rarely, sometimes, or often) | 20521<br>20522<br>20523<br>20524<br>20525 | *Davis et al., 2020* |
| Catastrophic trauma | At least one catastrophic event:<br>Victim of sexual assault<br>Victim of physically violent crime<br>Been in a serious accident believed to be life-threatening<br>Witnessed sudden violent death<br>Diagnosed with a life-threatening illness<br>Been involved in combat or exposed to war zones | 20526<br>20527<br>20528<br>20529<br>20530<br>20531 | *Davis et al., 2020* |
| Wellbeing | The sum of the following scores:<br>General happiness<br>Happiness with own health<br>Belief that life is meaningful | 20458<br>20459<br>20460 | *Davis et al., 2020* |
| Any distress | Reported functional impairment due to mental distress:<br>Ever sought/received help for mental distress<br>Mental distress prevented usual activities<br>Mental health problems diagnosed by a healthcare professional<br>Plus, a positive diagnosis for a specific condition (Depression ever, GAD ever, Addiction ever, Bipolar ever, Psychotic experiences, PTSD, Self-harm ever). | 20499<br>20500<br>20544 | *Davis et al., 2020* |

N-12, the Eysenck Neuroticism score; *PTSD*, post-traumatic stress disorder; *PCL-6*, PTSD Checklist; *PHQ-9*, Patient Health Questionnaire score; *PDS*, Probable Depression Status; *RDS-4*, Recent Depressive Symptoms; *AUDIT*, Alcohol Use Disorders Identification Test; *GAD-7*, Generalised Anxiety Disorder score; *GP*, general practitioner.

