## [Editor Report · eLife Assessment]

This **valuable** work advances our understanding of the relationship between multimodal magnetic resonance imaging (MRI) measures, cognition, and mental health. **Compelling** use of statistical learning techniques in UK Biobank data shows that 48% of the variance between an 11-task derived g-factor and imaging data can be explained. Overall, this paper contributes to the study of brain-behaviour relations and will be of interest for both its methods and its findings on how much variance in g can be explained.

---

## [Referee Report · Reviewer #1 (Public review)]

Summary:

The authors aimed to examine how the covariation between cognition (represented by a g-factor based on 12 features of 11 cognitive tasks) and mental health (represented by 133 diverse features) is reflected in MR-based neural markers of cognition, as measured through multimodal neuroimaging (structural, rsfMRI and diffusion MR). To integrate multiple neuroimaging phenotypes across MRI modalities the authors used a so-called a stacking approach, which employs two levels of machine learning. First, they build a predictive model from each neuroimaging phenotype to predict a target variable. Next, in the stacking level, they use predicted values (i.e., cognition predicted from each neuroimaging phenotype) from the first level as features to predict the target variable. To quantify the contribution of the neural indicators of cognition explaining the relationship between cognition and mental health, they conducted commonality analyses. Results showed that when they stacked neuroimaging phenotypes within dwMRI, rsMRI, and sMRI, they captured 25.5%, 29.8%, and 31.6% of the predictive relationship between cognition and mental health, respectively. By stacking all 72 neuroimaging phenotypes across three MRI modalities, they enhanced the explanation to 48%. Age and sex shared substantial overlapping variance with both mental health and neuroimaging in explaining cognition, accounting for 43% of the variance in the cognition-mental health relationship.

Strengths:

(1) Big study population (UK Biobank with 14000 subjects)

(2) Description of methods (including Figure 1) is helpful in understanding the approach

(3) Final manuscript improved after revision

Weaknesses:

(1) The relevance of the question is now better described, but the impact of the work is more of conceptual value than of direct clinical value.

(2) The discussion on the interpretation of the positive and negative PLRS loadings is now further explained, but remains a bit counterintuitive.

Note: the computational aspects of the methods fall beyond my expertise.

---

## [Referee Report · Reviewer #2 (Public review)]

Summary:

The goal of this manuscript was to examine whether neural indicators explain the relationship between cognition and mental health. The authors achieved this aim by showing that the combination of MRI markers better predicted the cognition-mental health covariation. I have reviewed the paper before and the authors addressed my comments very well.

Strengths:

Large sample (UK biobank data) and clear description of advanced analyses.

Weaknesses:

My main concern in my previous review was that it was not completely clear to me what it means to look at the overlap between cognition and mental health. The authors have addressed this in the current version.

---

## [Author Response]

The following is the authors’ response to the original reviews.

**Public Reviews:**

**Reviewer #1 (Public review):**
Summary:The authors aimed to examine how the covariation between cognition (represented by a *g*-factor based on 12 features of 11 cognitive tasks) and mental health (represented by 133 diverse features) is reflected in MR-based neural markers of cognition, as measured through multimodal neuroimaging (structural, rsfMRI, and diffusion MR). To integrate multiple neuroimaging phenotypes across MRI modalities, they used a so-called stacking approach, which employs two levels of machine learning. First, they built a predictive model from each neuroimaging phenotype to predict a target variable. Next, in the stacking level, they used predicted values (i.e., cognition predicted from each neuroimaging phenotype) from the first level as features to predict the target variable. To quantify the contribution of the neural indicators of cognition explaining the relationship between cognition and mental health, they conducted commonality analyses. Results showed that when they stacked neuroimaging phenotypes within dwMRI, rsMRI, and sMRI, they captured 25.5%, 29.8%, and 31.6% of the predictive relationship between cognition and mental health, respectively. By stacking all 72 neuroimaging phenotypes across three MRI modalities, they enhanced the explanation to 48%. Age and sex shared substantial overlapping variance with both mental health and neuroimaging in explaining cognition, accounting for 43% of the variance in the cognition-mental health relationship.Strengths:(1) A big study population (UK Biobank with 14000 subjects).(2) The description of the methods (including Figure 1) is helpful in understanding the approach.(3) This revised manuscript is much improved compared to the previous version.Weaknesses:(1) Although the background and reason for the study are better described in this version of the manuscript, the relevance of the question is, in my opinion, still questionable. The authors aimed to determine whether neural markers of cognition explain the covariance between cognition and mental health and which of the 72 MRI-based features contribute to explaining most of the covariance. I would like to invite the authors to make a stronger case for the relevance, keeping the clinical and scientific relevance in mind (what would you explain to the clinician, what would you explain to the people with lived experience, and how can this knowledge contribute to innovation in mental health care?).

Thank you for this insightful observation. We agree that establishing the real-world significance of fundamental research is paramount, and we have revised our manuscript to better articulate this relevance.

For clinicians, our work (a) corroborates the link between cognition and mental health, confirming the transdiagnostic role of cognition, and (b) demonstrates that current neuroimaging tools can capture the neurobiology underlying this relationship. These findings offer several implications for clinical practice. First, they support the development of interventions aimed at enhancing cognitive functioning as a pathway to improving mental health. Second, our work introduces neuroimaging as a potential tool for assessing the neurobiological basis of the cognition–mental health connection. With further research, clinicians may be able to use neuroimaging to track cognitive changes at the neural level, which could help monitor treatment efficacy for interventions (e.g., stimulant medications for ADHD) designed to boost cognitive functioning.

Following your suggestions, we have expanded the Discussion (Line 684) to include future directions and clinical perspectives on the findings.

Line 684: “Neuroimaging offers a unique window into the biological mechanisms underlying cognition–mental health overlap – insights unattainable from behavioural data alone. Our findings validate brain-based neural markers as a core unit of analysis for cognitive functioning, advancing mental health research through the lens of cognition. Beyond this conceptual contribution, the study has clinical implications. First, by demonstrating a transdiagnostic link between cognition and mental health, we support interventions that enhance cognition as a pathway to improving mental health. Second, we show neuroimaging as an effective tool for assessing the neurobiological basis of this link. Quantifying neuroimaging’s capacity to capture this relationship is essential for future research integrating imaging with cognitive testing to monitor treatment-related neural changes. Such work could enable personalised interventions, using neuroimaging to track cognitive changes and treatment efficacy (e.g., stimulant medications for ADHD) aimed at boosting cognitive functioning.”

(2) The discussion on the interpretation of the positive and negative PLRS loadings is not very convincing, and the findings are partly counterintuitive. For example (1) how to explain that distress has a positive loading and anxiety/trauma has a negative loading?; (2) how to explain that mental health features like wellbeing and happiness load in the same direction as psychosis and anxiety/trauma? From both a clinical and a neuroscientific perspective, this is hard to interpret.

Thank you for pointing this out. We appreciate your concern regarding the interpretation of positive and negative PLSR loadings. To clarify:

(1) The directions of PLSR loadings are broadly consistent with univariate correlations, suggesting that the somewhat counterintuitive relationships mentioned are shown even when we apply simply univariate correlations. PLSR extends beyond univariate approaches by modelling multivariate relationships across features and outcomes. It constructs new components – linear combinations of predictors – that simultaneously explain variance in the predictors and their covariance with the response.

(2) The positive loading of distress likely reflects cohort-specific questionnaire design in the UK Biobank, where feeling of distress was tied to seeking medical help. Individuals with higher cognition and socioeconomic status may be more likely to seek professional support, which explains the counterintuitive direction.

(3) The negative loadings of wellbeing and happiness may also reflect cohort-specific effects, such as older age, and align with prior work linking excessive optimism to poorer reasoning and cognitive performance. This suggests that realism or pessimism may sometimes be associated with better cognition, particularly in older adults.

These points are discussed in detail in the manuscript (Lines 493–545). We have emphasised that some of these findings may be cohort-specific and cited supporting literature, as seen below.

(1) How to explain that distress has a positive loading and anxiety/trauma has a negative loading?

Line 493: “The directions of PLSR loadings were broadly consistent with univariate correlations. PLSR extends beyond univariate approaches by modelling multivariate relationships across features and outcomes. Consistently, both univariate correlations and factor loadings derived from the PLSR model indicated that scores for mental distress, alcohol and cannabis use, and self-harm behaviours related positively, and the scores for anxiety, neurological and mental health diagnoses, unusual or psychotic experiences, happiness and subjective well-being, and negative traumatic events related negatively to the *g*-factor. Positive PLSR loadings of features related to mental distress may indicate greater susceptibility to or exaggerated perception of stressful events, psychological overexcitability, and predisposition to rumination in people with higher cognition [72]. On the other hand, these findings may be specific to the UK Biobank cohort and the way the questions for this mental health category were constructed. In particular, to evaluate mental distress, the UK Biobank questionnaire asked whether an individual sought or received medical help for or suffered from mental distress. In this regard, the estimate for mental distress may be more indicative of whether an individual experiencing mental distress had an opportunity or aspiration to visit a doctor and seek professional help [73]. Thus, people with better cognitive abilities and also with a higher socioeconomic status may indeed be more likely to seek professional help.”

Line 529: “Consistent with previous studies, we showed that anxiety and negative traumatic experiences were inversely associated with cognitive abilities [90–93]. Anxiety may be linked to poorer cognitive performance via reduced working memory capacity, increased focus on negative thoughts, and attentional bias to threatening stimuli that hinder the allocation of cognitive resources to a current task [94–96]. Individuals with PTSD consistently showed impaired verbal and working memory, visual attention, inhibitory function, task switching, cognitive flexibility, and cognitive control [97–100]. Exposure to traumatic events that did not reach the PTSD threshold was also linked to impaired cognition. For example, childhood trauma is associated with worse performance in processing speed, attention, and executive function tasks in adulthood, and age at a first traumatic event is predictive of the rate of executive function decline in midlife [101,102]. In the UK Biobank cohort, adverse life events have been linked to lower cognitive flexibility, partially via depression level [103].”

(2) How to explain that mental health features like wellbeing and happiness load in the same direction as psychosis and anxiety/trauma?

Line 545: “Finally, both negative PLSR loadings and corresponding univariate correlations for features related to happiness and subjective well-being may be specific to the study cohort, as these findings do not agree with some previous research [107–109]. On the other hand, our results agree with the study linking excessive optimism or optimistic thinking to lower cognitive performance in memory, verbal fluency, fluid intelligence, and numerical reasoning tasks, and suggesting that pessimism or realism indicates better cognition [110]. The concept of realism/optimism as indicators of cognition is a plausible explanation for a negative association between the gfactor and friendship satisfaction, as well as a negative PLSR loading of feelings that life is meaningful, especially in older adults who tend to reflect more on the meaning of life [111]. The latter is supported by the study showing a negative association between cognitive function and the search for the meaning of life and a change in the pattern of this relationship after the age of 60 [112]. Finally, a UK Biobank study found a positive association of happiness with speed and visuospatial memory but a negative relationship with reasoning ability [113].”

(3) The analysis plan has not been preregistered (e.g. at OSF).Note: the computational aspects of the methods fall beyond my expertise.

Thank you for pointing this out. We acknowledge that the analysis plan was not preregistered, as our approach was primarily data‑driven rather than hypothesis‑driven. We essentially applied the machine learning approach to quantify the strength of the cognition-mental health relationship in relation to neuroimaging. To ensure transparency and reproducibility, we have made all analysis code and intermediate outputs publicly available on our GitHub repository (https://github.com/HAM-lab-Otago-University/UKBiobank/) within the constraints of UK Biobank’s ethical policy and provided a detailed description of each methodological step in the Supplementary Materials.

**Reviewer #2 (Public review):**
Summary:The goal of this manuscript was to examine whether neural indicators explain the relationship between cognition and mental health. The authors achieved this aim by showing that the combination of MRI markers better predicted the cognition-mental health covariation.Strengths:The evidence supporting the conclusions is compelling. There is a large sample (UK biobank data) and a clear description of advanced analyses.Weaknesses:In the previous version of the paper, it was not completely clear what it means to look at the overlap between cognition and mental health. The authors have addressed this in the current version.

Thank you for your positive feedback and for recognizing the strengths of our work. We appreciate your comments and are happy that the revisions addressed your concerns.